# Eukaryotic-like gephyrin and cognate membrane receptor coordinate corynebacterial cell division and polar elongation

Mariano Martinez [1,6], Julienne Petit [1,6], Alejandro Leyva [2,6], Adrià Sogues [1,5], Daniela Megrian [1], Azalia Rodriguez[2], Quentin Gaday[1], Mathildeb Ben Assaya[1], Maria Magdalena Portela[2], Ahmed Haouz [3], Adrien Ducret [4], Christophe Grangeasse [4], Pedro M. Alzari [1], Rosario Durán [2]✉ & Anne Marie Wehenkel [1]✉

The order Corynebacteriales includes major industrial and pathogenic Actinobacteria such as *Corynebacterium glutamicum* or *Mycobacterium tuberculosis*. These bacteria have multi-layered cell walls composed of the mycolyl-arabinogalactan-peptidoglycan complex and a polar growth mode, thus requiring tight coordination between the septal divisome, organized around the tubulin-like protein FtsZ, and the polar elongasome, assembled around the coiled-coil protein Wag31. Here, using *C. glutamicum*, we report the discovery of two divisome members: a gephyrin-like repurposed molybdotransferase (Glp) and its membrane receptor (GlpR). Our results show how cell cycle progression requires interplay between Glp/GlpR, FtsZ and Wag31, showcasing a crucial crosstalk between the divisome and elongasome machineries that might be targeted for anti-mycobacterial drug discovery. Further, our work reveals that Corynebacteriales have evolved a protein scaffold to control cell division and morphogenesis, similar to the gephyrin/GlyR system that mediates synaptic signalling in higher eukaryotes through network organization of membrane receptors and the microtubule cytoskeleton.

Cell division is central to bacterial physiology. Since the seminal work of Francois Jacob in 1968 on the *filamentation temperature-sensitive* (*fts*) genes in *Escherichia coli*, which led to the discovery of the tubulin-like bacterial cytoskeleton protein FtsZ[1], a few well-studied model systems set the basis for our current knowledge of cell division at the molecular level. In this process, FtsZ regulates—through GTP-dependent polymerization—the assembly of the cell division machinery (the divisome) at the site of septation and governs the ordered assembly

[1]Structural Microbiology Unit, Institut Pasteur, CNRS UMR 3528, Université Paris Cité, Paris, France. [2]Analytical Biochemistry and Proteomics Unit, Institut Pasteur de Montevideo, Instituto de Investigaciones Biológicas Clemente Estable, Montevideo, Uruguay. [3]Plate-forme de cristallographie, C2RT-Institut Pasteur, CNRS UMR 3528, Université Paris Cité, Paris, France. [4]Molecular Microbiology and Structural Biochemistry, CNRS UMR 5086, Université de Lyon, Lyon, France. [5]Present address: Structural and Molecular Microbiology, VIB-VUB Center for Structural Biology, VIB, Vrije Universiteit Brussel, Brussels, Belgium. [6]These authors contributed equally: Mariano Martinez, Julienne Petit, Alejandro Leyva. ✉e-mail: duran@pasteur.edu.uy; anne-marie.wehenkel@pasteur.fr

of the cell wall biosynthetic machinery[2]. However, while many cell division genes and interaction networks were identified in model organisms, the detailed molecular mechanisms underlying bacterial cell division remain enigmatic, notably because of the diversity of species-specific adaptations[2–4]. In Actinobacteria, a large phylum that includes important human pathogens such as *Mycobacterium tuberculosis* and *Corynebacterium diphtheriae*, many of the well-studied components of the divisome from *E. coli* or *Bacillus subtilis* are missing from the genomes[5]. This is especially the case for several FtsZ regulatory proteins, including FtsA, EzrA and ZipA. So far, only the essential membrane anchor SepF has been unequivocally identified as a direct interactor with the C-terminal domain of FtsZ (FtsZ$_{CTD}$)[6,7]. This apparent lack of divisome components in Corynebacteriales is particularly intriguing, as these polar-growing bacteria need to coordinate the mid-cell division and elongation machineries at a precise moment of the cell cycle when the septum becomes a new pole[8]. Furthermore, their complex multi-layered cell wall formed by peptidoglycan, arabinogalactan and the mycolate outer membrane[9,10] needs to be fully assembled before cytokinesis[11].

How the polar elongasome is assembled remains to be elucidated. In laterally elongating bacteria such as *E. coli* or *B. subtilis*, the actin-like MreB scaffold organizes the elongasome[12–14]. In Corynebacteriales, MreB is absent from the genomes. Instead, the cytoskeletal DivIVA homologue Wag31, a coiled-coil scaffolding protein with an N-terminal membrane-binding domain, is essential for assembling the polar elongasome and preserving the rod-shaped morphology of Corynebacteriales[15–17]. Several proteins have been described as putative Wag31 interactors, mainly from genetic and cellular experiments, such as ParA/B[18,19], CwsA[20], RodA[16] or MksG[21]. Wag31 has a subpolar localization and, concomitant with or soon after septum formation, migrates to the cell division site at mid-cell to eventually assemble the daughter cell elongasome at the new cell pole[22]. The old pole grows faster than the new pole, suggesting that full maturation of polar and subpolar assemblies occurs over time and is divisome-independent[23,24]. How division and elongation processes are related in space and time and to what extent Wag31 is directly involved in protein–protein interactions with other components of these machineries is unknown.

Here we report the discovery of two members of the corynebacterial divisome and the dissection of a regulatory cell cycle network that directly links FtsZ to Wag31. We show that this link is mediated by a gephyrin-like protein (Glp) and its membrane receptor (GlpR). Mammalian gephyrin is a moonlighting protein that plays an essential role in synaptic signalling via clustering of the glycine receptor GlyR. Like gephyrin, Glp has undergone evolutionary repurposing in Actinobacteria to bind FtsZ. Our studies show that Glp and GlpR form a tight complex that is part of the early divisome, where Glp and GlpR directly bind FtsZ and Wag31, respectively, placing the Glp–GlpR complex at the centre of the divisome–elongasome transition.

## Results

### Glp is a divisome component in Actinobacteria

To discover missing players in corynebacterial cell division, we used mass-spectrometry-based interactomics, starting with the FtsZ membrane anchor SepF as bait for co-immunoprecipitation (co-IP) studies. *Corynebacterium glutamicum* (*Cglu*) cultures were cross-linked during exponential growth to stabilize interactions that are highly dynamic or that depend on spatial cues such as the inner membrane or the FtsZ polymerization status. We conducted co-IPs using the mScarlet fluorescent protein tag as bait in *Cglu* strains expressing either SepF-mScarlet or mScarlet. In addition, we used anti-SepF antibodies in both the untransformed *Cglu* and the SepF-mScarlet strain. Twenty proteins representing putative direct or indirect SepF interactors were exclusively detected or statistically enriched from *Cglu* when compared with the control, and 22 and 98 proteins were recovered from the *Cglu*_SepF-mScarlet strain using anti-SepF and anti-mScarlet antibodies

(Extended Data Fig. 1a and Supplementary Table 1a). Common to all IPs, 11 proteins with a quantifiable enrichment factor represent the SepF core interactome (Fig. 1a). As expected, this core interactome includes FtsZ as well as SepH, a recently identified FtsZ interactor in Actinobacteria[25], but the most enriched protein compared with the *Cglu* proteome is *Cgl0883*. This top interactor appeared consistently in all replicates and was named Glp (explained below). In *Cglu*, Glp is annotated as one of three molybdopterin molybdotransferase MoeA enzymes (EC 2.10.1.1) that incorporate the molybdenum metal into the molybdopterin (MPT) precursor to form the Moco co-factor used by molybdoenzymes to catalyse redox reactions[26].

A Glp knockout strain (*Cglu*_Δ*glp*, Extended Data Fig. 1b) was viable but displayed a strong cell division phenotype, with elongated, wider cells and multiple septa (Fig. 1b,c), suggesting a delay in the final steps of cell division. The *Cglu*_Δ*glp* strain was sensitive to the anti-tuberculosis drug ethambutol (Fig. 1d), a further indication of Glp involvement in cell division, as sublethal concentrations of ethambutol have been used to identify genes required for cell division[27]. Both the multi-septate and ethambutol-sensitive phenotypes were restored to wild type (WT) when Glp or mNeon-Glp were expressed from a plasmid under the control of P$_{gntK}$ (ref. 28) (Fig. 1b–d). Fluorescently labelled mNeon-Glp localized to mid-cell before septum formation, placing it with the early arrivers to the site of cell division (Fig. 1e). In contrast, the two paralogues of Glp in the *Cglu* genome, MoeA1/*Cgl0212* and MoeA3/*Cgl1196*, displayed a cytoplasmic distribution (Extended Data Fig. 1c,d), suggesting that only Glp evolved specific functions related to the divisome.

### Glp is a gephyrin-like protein that interacts with FtsZ

The eukaryotic gephyrin is described as a moonlighting enzyme originally identified as a glycine receptor-associated protein in neurons[29–31]. It was later found that the E-domain of gephyrin corresponds to MoeA and functions as a Moco biosynthetic enzyme[32,33]. Full-length gephyrin, which has an additional MogA domain, acts as a scaffold through oligomerization[34] and transiently clusters and stabilizes glycine (Gly) and GABAA receptors at the post-synapse of the mammalian brain[33,35]. It is thus tempting to speculate that Glp could also form protein networks in bacteria upon association with cell division proteins. Since there is evidence for a physical linkage between gephyrin, GlyR and microtubules[30], we tested whether Glp septum localization could be accounted for by a direct interaction with the bacterial tubulin homologue FtsZ. Our interactomics data were consistent with this hypothesis because, when using a SepF mutant unable to bind FtsZ as bait, we saw a significant decrease in Glp binding (Fig. 2a and Supplementary Table 1b), suggesting that the observed SepF–Glp interaction was indirect and occurred via FtsZ. This was further confirmed in vitro with purified proteins. We could not detect direct binding between SepF and Glp, but we could measure a direct interaction between Glp and FtsZ with an apparent $K_d$ of 4.7 ± 0.77 μM as determined by biolayer interferometry (BLI, Fig. 2b). The interaction is stronger with polymerized FtsZ in the presence of GTP (Extended Data Fig. 2a) and mediated by the conserved FtsZ$_{CTD}$ (Extended Data Fig. 2b).

We crystallized Glp alone and in complex with the 10-residue peptide FtsZ$_{CTD}$. The apo-structure was solved at 2.1 Å resolution and the structure of the complex (Fig. 2c,d) was solved at 2.7 Å resolution (Supplementary Table 2). The overall architecture of the Glp dimer and the monomer organization into four structural domains (I–IV) are similar to those described for *E. coli* MoeA[34] or the E-domain of gephyrin[36]. However, a pronounced hinge motion around the segments connecting structural domains I and III leads to an open form of the Glp homodimer compared with the closed form of *E. coli* MoeA (Fig. 2e). This conformational change, also seen in the absence of ligand (Extended Data Fig. 2c), generates the FtsZ-binding site within the central Glp dimer interface, far from the putative Mo-active site (Fig. 2c). The FtsZ$_{CTD}$ is well defined in the electron density (Extended Data Fig. 2d)

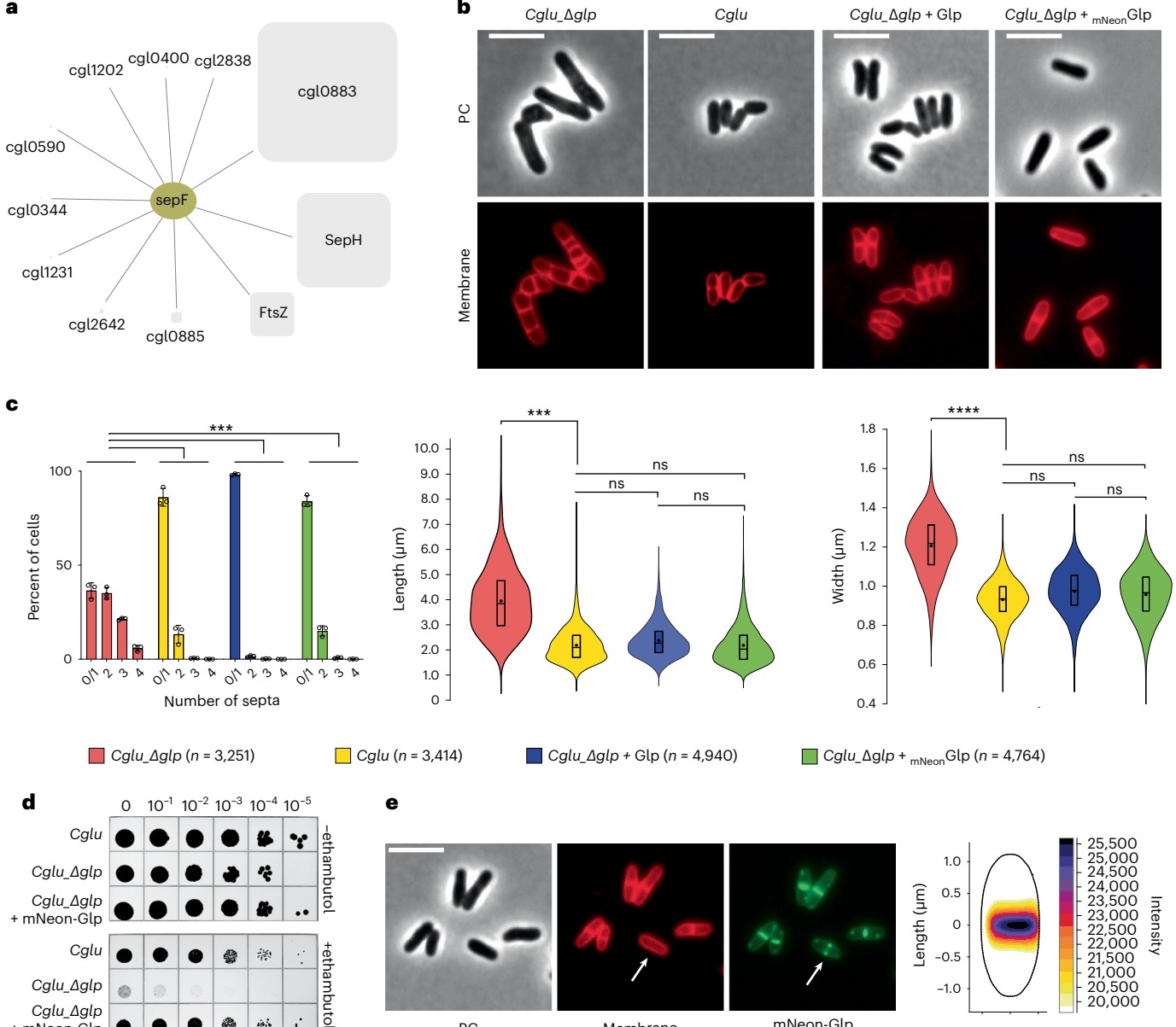

**Fig. 1 | Identification of Glp as a member of the corynebacterial divisome.**
**a**, The core interactome of SepF, including proteins recovered from three independent co-IP experiments using different strains/antibodies: *Cglu*/α-SepF, *Cglu*_SepF-Scarlet/α-SepF and *Cglu*_SepF-Scarlet/α-Scarlet. The square size for each interactor is proportional to its enrichment in the interactome compared to the proteome (Supplementary Table 1), and the lines indicate either direct or indirect interactors. **b**, Glp depletion and complementation. Representative images in phase contrast (PC) and membrane staining for indicated strains. **c**, Left: frequency histogram showing the number of septa per cell for the different strains, calculated from *n* cells imaged (indicated in the figure) from three independent experiments for each strain (*Cglu_Δglp*, *n* = 873, 1,538 and 840; *Cglu*, *n* = 718, 1,468 and 1,223; *Cglu_Δglp* + *Glp*, *n* = 2,465, 1,169 and 1,297; *Cglu_Δglp* + *mNeon-Glp*, *n* = 1,641, 1,311 and 1,801); open circles represent the corresponding data points; mean ± s.d.; Cohen's *d* (see Methods for interpretation of values), from top to bottom: (***d* = 1.57, *P* = 0), (***d* = 1.84, *P* = 0), (***d* = 1.6, *P* = 0). Middle

and right: violin plots showing the distribution of cell length (Cohen's *d*, from top to bottom: (***d* = 1.76, *P* ~ 0), ($^{ns}d$ = 0, *P* = 0.95), ($^{ns}d$ = 0.29, *P* = 3.78 × 10⁻³⁷), ($^{ns}d$ = 0.27, *P* = 2.59 × 10⁻³⁹)) and cell width (Cohen's *d*: (****d* = 2.21, *P* ~ 0), ($^{ns}d$ = 0.25, *P* = 4.13 × 10⁻²⁵), ($^{ns}d$ = 0.38, *P* = 9.65 × 10⁻⁷⁴), ($^{ns}d$ = 0.08, *P* = 1.74 × 10⁻¹²)); the box indicates the 25th to the 75th percentile, the mean and the median are indicated with a dot and a line in the box, respectively. **d**, Ethambutol sensitivity assay. BHI overnight cultures of *Cglu* and *Cglu_Δglp* complemented with the empty plasmid or mNeon-Glp were normalized to an OD₆₀₀ of 0.5, serially diluted 10-fold and spotted onto BHI agar medium with or without 1 μg ml⁻¹ ethambutol. **e**, Left: localization of mNeon-Glp in *Cglu*. Representative images in PC, membrane staining and mNeon-Glp fluorescent signals. The arrow indicates the Glp localization before septum formation. Right: heat map representing the localization pattern of mNeon-Glp; 3,879 cells were analysed, from triplicate experiments. Scale bars, 5 μm.

and interacts primarily with a protruding β-hairpin in Glp domain IV (Fig. 2d). Interestingly, the known binding sites of GlyR on mammalian gephyrin and FtsZ_CTD on Glp both map to structural domain IV, although not to the same binding site (Extended Data Fig. 2e). To validate the Glp–FtsZ interaction observed in the crystal, we produced a deletion

mutant of the entire FtsZ-binding loop between Met361 and Leu370 (GlpΔ_loop). The purified mutant protein was correctly folded (Extended Data Fig. 2f) but was unable to interact with FtsZ in vitro (Extended Data Fig. 2g). Complementation of the *Cglu_Δglp* strain with mNeon-Glp_Δloop failed to restore the septal localization or the wild-type morphology

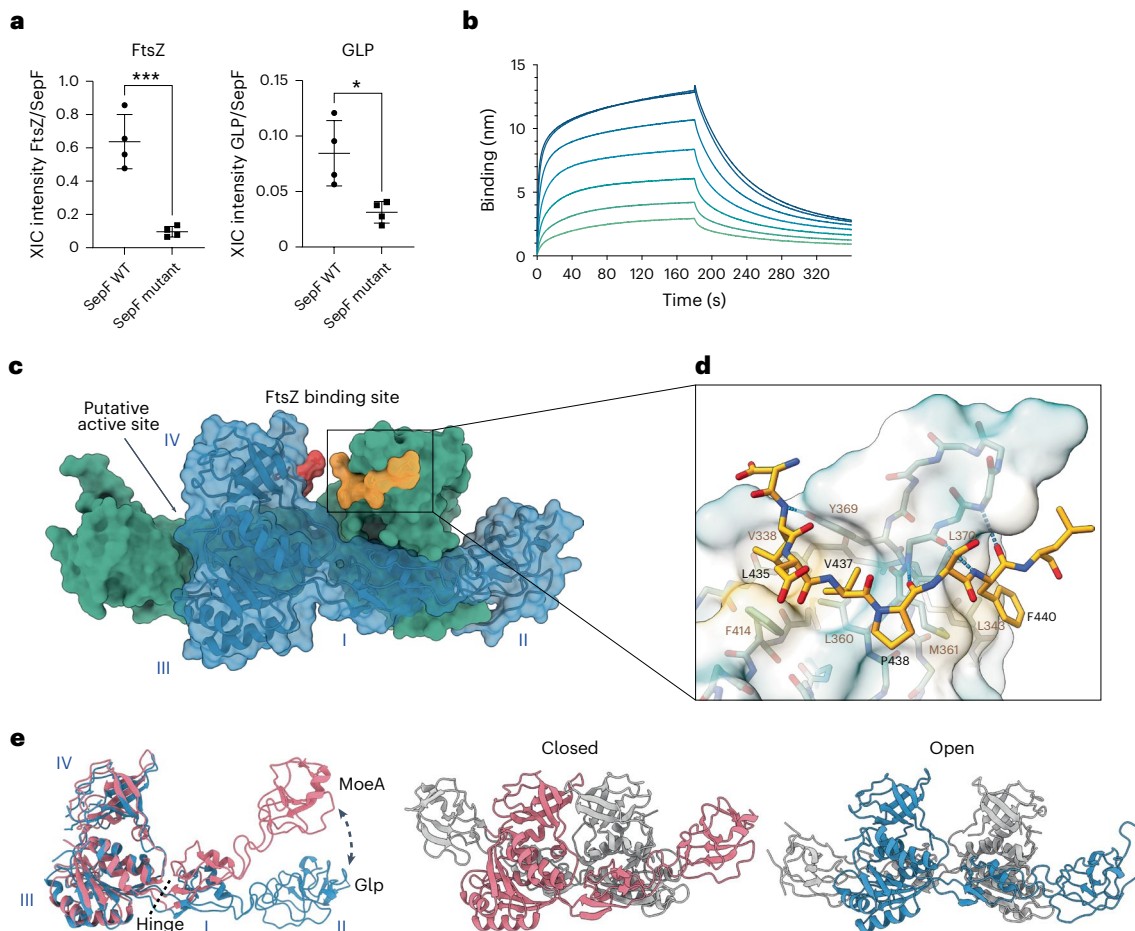

**Fig. 2 | Glp–FtsZ interaction. a**, Comparison of the recovery of FtsZ and Glp in co-IP (α-Scarlet) of *Cglu*_SepF-Scarlet and the mutant unable to bind FtsZ (SepF$_{K125E/F131A}$-Scarlet). Each point corresponds to the normalized XIC intensity in each replicate for each condition, calculated as described in Methods; *n* = 4 biologically independent samples per condition; mean ± s.d. Statistical analysis was performed using unpaired two-sided Student's *t*-test. FtsZ fold change (FC) = 6.61 (*P* = 0.0006); Glp FC = 2.70 (*P* = 0.014). See Supplementary Table 1b for corresponding analysis. **b**, BLI sensorgrams of Glp binding to immobilized SUMO-FtsZ. Glp concentrations range from 80 µM (dark blue) to 1.25 µM (light green) in 2-fold dilutions. **c**, Crystal structure of the Glp homodimer (blue and green) in complex with FtsZ$_{CTD}$ (yellow and red). The Glp monomer is composed of 4 structural domains (labelled I–IV): domain I (residues 20–45 and 146–181), domain II (residues 46–145), domain III (residues 1–19 and 182–331) and domain IV (residues 332–417). The location of the putative active site at the distal dimer interface is indicated. **d**, Detailed view of Glp–FtsZ interactions. The peptide

adopts a linear extended conformation, with a central kink promoted by the presence of Pro438. The C-terminal half of the peptide backbone runs roughly parallel to the Glp β-strand 360–363 and is stabilized by three intermolecular hydrogen bonds between main-chain atoms (N$_{R362}$-O$_{P438}$, O$_{R362}$-N$_{F440}$ and N$_{A364}$-O$_{F440}$) and by hydrophobic interactions (FtsZ Phe440 with Glp Leu343, Met361 and Leu370). On the N-terminal half, the side chains of FtsZ residues Leu435 and Val437 are anchored in a hydrophobic pocket defined by Glp residues Val338, Leu360, Tyr369 and Phe414. Residues involved in protein–protein interactions are labelled; the molecular surface of Glp shows hydrophobicity (yellow, hydrophobic; green, hydrophilic); intermolecular hydrogen bonds are shown as blue dotted lines. **e**, Left: the superposition of the monomers from Glp (blue) and MoeA from *E. coli* (pink, pdb 1g8l) reveals a pronounced conformational change from a hinge region at the interface between domain I and III. This change leads to a central open (right; Glp, blue) or closed (middle; MoeA, pink) conformation in the respective homodimers.

(Extended Data Fig. 2h,i), stressing the physiological relevance of the crystallographic Glp–FtsZ complex.

## GlpR is a membrane receptor for Glp

To further investigate Glp function, we performed the reverse interactome, this time using Glp as bait (Supplementary Table 1c). Besides recovering FtsZ and SepF as expected, we identified *Cgl0885*, a membrane protein of unknown function (named GlpR hereafter) that was already present among the top SepF interactors (Fig. 1a and Supplementary Table 1a). GlpR is an integral membrane protein with 3 predicted transmembrane (TM) helices and two cytoplasmic, oppositely charged intrinsically disordered regions (IDRs) (Fig. 3a). To determine whether there is a direct interaction between Glp and GlpR, we produced the recombinant proteins and assessed their interaction in vitro. The two proteins form a high-affinity complex (apparent *K*$_d$

of 5.5 ± 0.75 nM, Fig. 3b), indicating that GlpR might function as a membrane receptor for Glp. Glp$_{Δloop}$ was unable to interact with GlpR (Extended Data Fig. 3), suggesting that both FtsZ and GlpR bind to an overlapping region at the centre of the Glp dimer. In cellular fractionation assays, Glp is found in the membrane fraction despite not having any membrane anchoring domains (Fig. 3c). Glp localization to the membrane is reduced, but not abolished, in the *Cglu_Δglpr* depletion strain, suggesting the probable contribution of other proteins (for example, FtsZ) to the membrane partitioning and septum localization of Glp. This agrees with the observation that Glp can still localize to the septum in the absence of GlpR (Fig. 3d). Taken together our results show that Corynebacteriales have evolved a gephyrin/ GlyR-like system involved in cell division, prompting us to name the genes *Cgl0883* as Glp (for gephyrin-like protein) and *Cgl0885* as GlpR (for Glp receptor).

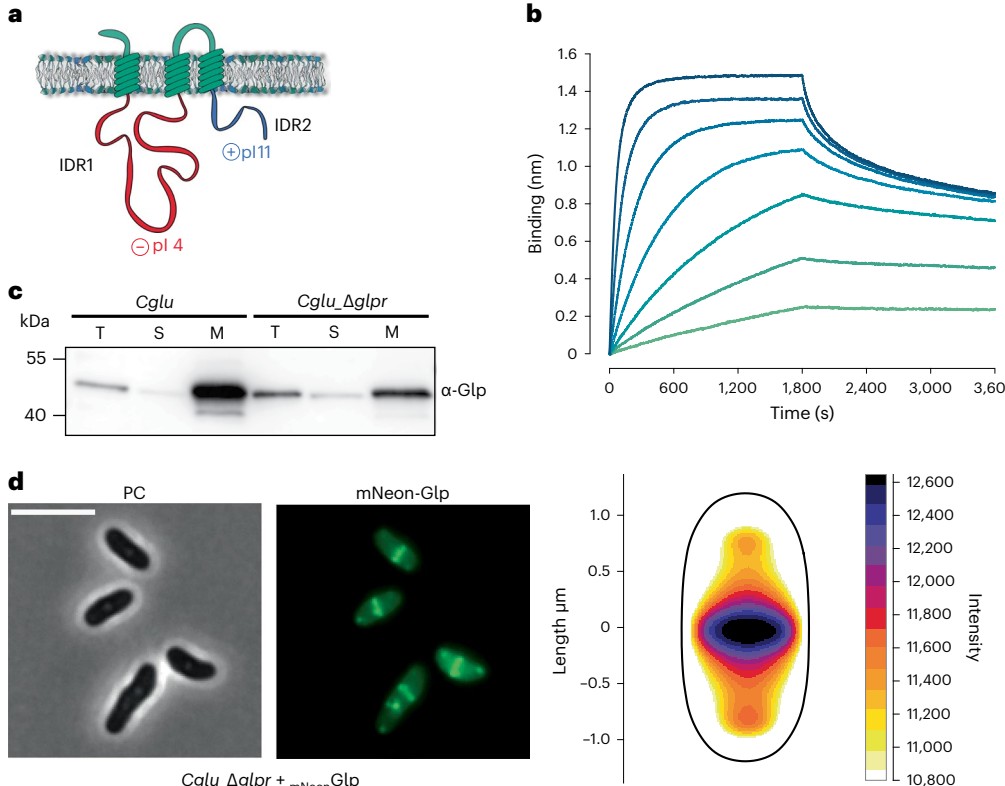

**Fig. 3 | Identification of GlpR as a membrane receptor for Glp. a**, Schematic representation of GlpR, with 3 transmembrane segments and the intrinsically disordered regions IDR1 (in red, residues 27–218, theoretical isoelectric points (pI) of 4.05) and IDR2 (in blue, residues 263–340, pI of 10.87). **b**, BLI sensorgrams of Glp binding to immobilized GlpR. Glp concentrations range from 100 nM (dark blue) to 1.56 nM (light green) in 2-fold dilutions. **c**, Cell fractionation and subcellular localization of Glp. Total (T), soluble (S) and membrane (M) fractions of *Cglu* or *Cglu_Δglpr* strains were obtained by differential centrifugation and analysed by western blot using an α-Glp antibody. **d**, Left: localization of mNeon-Glp in *Cglu_Δglpr*. Representative image in PC and mNeon-Glp fluorescent signal. Right: heat map representing the localization pattern of mNeon-Glp; 5,963 cells were analysed, from triplicate experiments. Scale bars, 5 μm.

## GlpR links the mid-cell divisome to the future polar elongasome

Unlike the *Cglu_Δglp* strain, depletion of GlpR in the *Cglu_Δglpr* strain (Extended Data Fig. 4a) did not show a significant morphological phenotype (Extended Data Fig. 4b) or sensitivity to ethambutol (Extended Data Fig. 4c), possibly due to functional redundancy of yet to be identified divisome members. At low levels of expression, GlpR-mNeon localized to the septum in *Cglu* and *Cglu_Δglpr* (Fig. 4a), and the cells displayed a mostly normal morphology (Fig. 4b). However, higher levels of GlpR-mNeon expression led to a strong morphotype characterized by the delocalization of the elongasome as revealed by aberrant pole formation along the lateral walls (branching, Fig. 4c). The cell projected surface area was significantly increased in *Cglu_Δglpr* + GlpR-mNeon when compared with *Cglu* (Fig. 4d). The observed phenotype is likely due to steric hindrance induced by mNeon, as the untagged overexpression of full-length GlpR or GlpR lacking the C-terminal IDR (GlpR_{ΔIDR2}) does not lead to branching (Fig. 4c–e). In fact, the lack of the GlpR-IDR2 domain partially phenocopies the *Cglu_Δglp* (Fig. 4e,f), suggesting that IDR2 is at least in part responsible for correct functioning of Glp.

In naturally branching actinomycetes such as *Streptomyces*, apical growth is directed by the essential coiled-coil protein DivIVA, which marks the hyphal site[37,38]. Similarly, Wag31, the Corynebacteriales homologue of *Streptomyces* DivIVA, specifically marks the sites of growth and its dysregulation results in polar growth from incorrect sites in *M. smegmatis*[39,40], suggesting that Wag31 delocalization is linked to the branching phenotype of the GlpR-mNeon overexpression strain (Fig. 4c). The hypothesis that the Glp–GlpR complex might exert a regulatory role on early elongasome assembly and localization by acting on Wag31 was further supported by IP experiments showing complex formation in vivo between Wag31 and GlpR. Pulling on GlpR with an anti-GlpR antibody showed the co-elution of the two proteins in *Cglu* (Fig. 4g). This co-elution was reduced in *Cglu_Δglp* and was not seen in *Cglu_Δglpr*. Moreover, quantitative analysis of the MS experiments revealed that Wag31 was not only systematically enriched in the Glp interactome (Supplementary Table 1c), but that it was also significantly decreased in the Glp interactome in the *Cglu_Δglpr* background (Fig. 4h). To seek direct biochemical evidence for in vitro interaction, we purified the full-length proteins (Glp, GlpR and Wag31) as well as the N-terminal DivIVA domain of Wag31 (Wag31_{1–61}). We were unable to detect any interaction between Glp and Wag31 under the conditions tested. In contrast, GlpR did bind both full-length Wag31 as well as Wag31_{1–61}, with apparent $K_d$ values of 43.4 ± 0.16 μM and 14.9 ± 1.4 μM, respectively (Fig. 4i and Extended Data Fig. 5), demonstrating that Wag31–GlpR complex formation is mediated at least in part through the N-terminal DivIVA domain. Taken together, the above data suggest that the FtsZ-associated Glp–GlpR complex regulates early elongasome assembly at mid-cell via direct interaction with Wag31, the scaffolding protein of the elongasome.

## Glp is a repurposed protein that co-evolved with GlpR

Most actinobacterial genomes contain at least two copies of MoeA. In the phylogeny of these homologues, we identified a monophyletic clade that contains *Cglu* Glp (Fig. 5a) and whose sequences are distinguished by two conserved proline-rich (pro-rich) regions (Fig. 5b). These regions correspond to the linkers connecting structural domains I and III, which differ between Glp and *E. coli* MoeA and

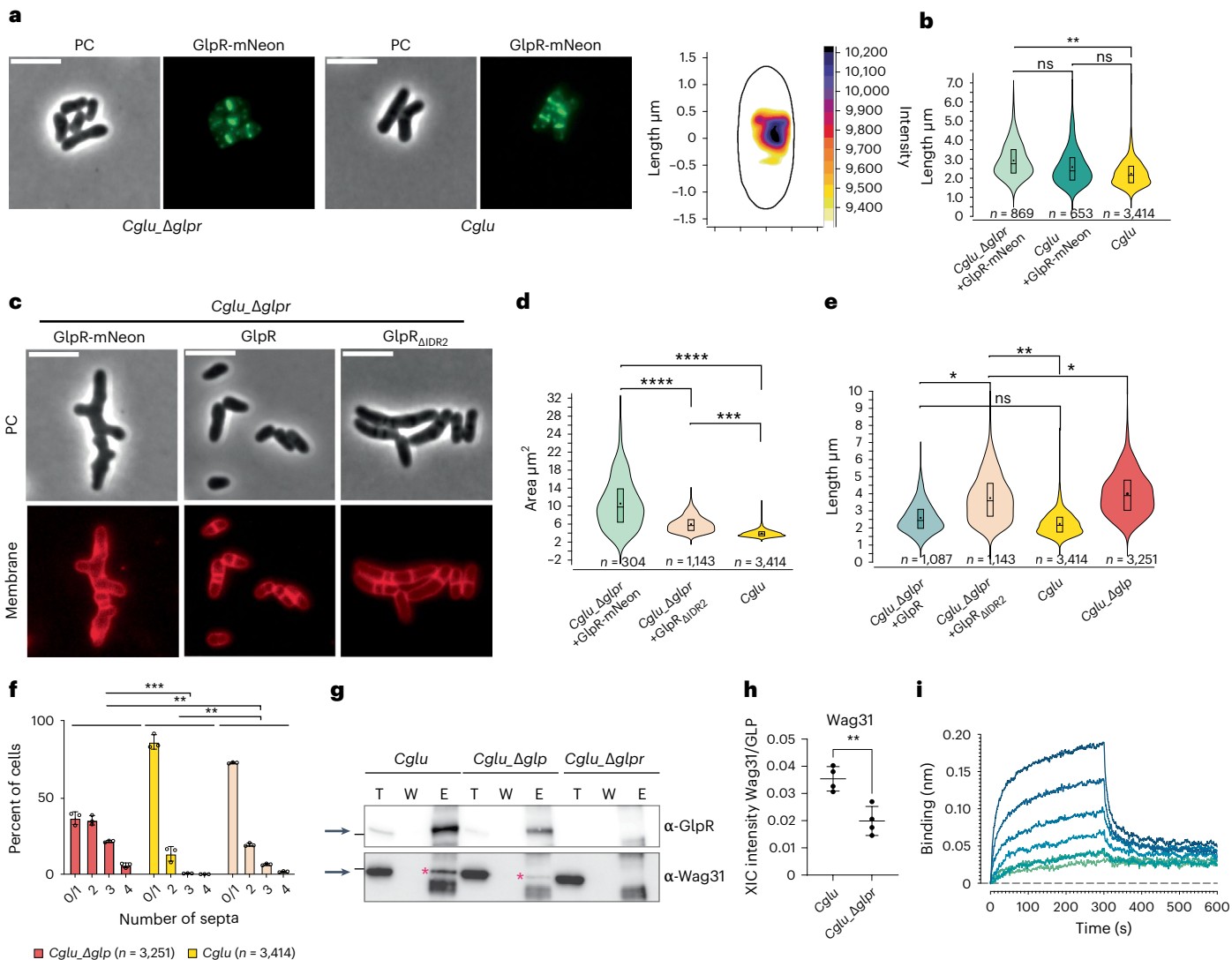

**Fig. 4 | GlpR links the mid-cell divisome with the future polar elongasome via Wag31. a**, Left: representative images of GlpR-mNeon expressed in *Cglu* and *Cglu_Δglpr*. Right: heat map of the localization of GlpR-mNeon in *Cglu* (*n* = 111). **b**, Violin plots (cell length). Cohen's *d*, from top to bottom: (**\*\****d* = 0.98, *P* = 2.88 × 10⁻⁸⁵), (ⁿˢ*d* = 0.49, *P* = 6.34 × 10⁻¹⁹), (ⁿˢ*d* = 0.39, *P* = 2.78 × 10⁻¹³). The expression levels of GlpR-mNeon are shown in Extended Data Fig. 4d. **c**, Representative images of *Cglu_Δglpr* complementation (overexpression conditions). **d**, Violin plots (cell surface areas). Cohen's *d*: (**\*\*\*\****d* = 3.95, *P* = 7.34 × 10⁻⁶³), (**\*\*\*\****d* = 2.35, *P* = 1.01 × 10⁻⁴⁶), (**\*\*\****d* = 1.4, *P* = 5.37 × 10⁻³¹⁹). The expression levels of GlpR-mNeon are shown in Extended Data Fig. 4e. **e**, Violin plots of cell length. Cohen's *d*: (**\*\****d* = 1.17, *P* = 7.22 × 10⁻²⁵⁵), (\**d* = 1.70, *P* = 6.20 × 10⁻¹³⁸), (\**d* = 0.61, *P* = 3.07 × 10⁻⁶⁰), (ⁿˢ*d* = 0.49, *P* = 8.12 × 10⁻³⁴). Boxes indicate the 25th to the 75th percentile, mean and median indicated with a dot and a line, respectively, in the box. Number of cells used (*n*) below the violin representation corresponds to triplicates. **f**, Histogram of number of septa per

cell, calculated from *n* cells from 3 independent experiments (*Cglu_Δglp*, *n* = 873, 1,538 and 840; *Cglu*, *n* = 718, 1,468 and 1,223; *Cglu_Δglpr* + GlpR_ΔIDR2, *n* = 451, 737 and 841); open circles represent the corresponding data points; mean ± s.d.; Cohen's *d*, from top to bottom: (**\*\*\****d* = 1.60, *P* = 0), (**\*\****d* = 0.80, *P* = 3.70 × 10⁻¹⁶⁹), (**\*\****d* = 0.79, *P* = 8.20 × 10⁻¹⁴⁷). **g**, Co-IP of GlpR-Wag31 for indicated strains using GlpR as bait. Total (T), wash (W) and elution (E) fractions were analysed by western blot using α-GlpR and α-Wag31 antibodies. Arrows indicate GlpR (top) and Wag31 (bottom) and Wag31 is additionally highlighted by a red *. The black bar corresponds to the 55 kDa molecular weight marker. **h**, Wag31 recovery in co-IPs of Glp from indicated strains. Each point corresponds to the normalized XIC intensity in each biologically independent replicate (*n* = 4) for each condition; mean ± s.d. Wag31 FC = 1.78 (*P* = 0.004). Statistical analysis was performed using a two-sided unpaired Student's *t*-test. **i**, BLI sensorgrams of Wag31 binding to GlpR. Wag31 concentrations: 150 μM (dark blue) to 2.3 μM (light green) in 2-fold dilutions.

are responsible for the hinge motion that generates the FtsZ-binding site in the former (Fig. 2e). Together, the two pro-rich regions and the FtsZ-binding loop can discriminate Glp from non-Glp MoeA homologues (Fig. 5b) and therefore represent a molecular signature of the functional repurposing for specific divisome functions. These results demonstrate that Glp has evolved to bind FtsZ and is recruited to the division site by the direct interaction of domain IV with the conserved C-terminal domain of FtsZ. Interestingly, synteny analysis in Actinobacteria revealed that both *glp* and *glpr* genes co-localize in

the genome (Fig. 5a and Extended Data Fig. 6a) and that their genomic context is well conserved. Moreover, when present, genes *glpr* and *glp* co-occur (Fig. 5c and Extended Data Fig. 7), suggesting a common evolutionary history.

The *glp* clade contains sequences mostly restricted to Actinomycetes, the largest class of Actinobacteria that includes the order Corynebacteriales (Fig. 5a), and the topology of the clade resembles that of Actinomycetes species (Fig. 5c). This suggests that *glp* was obtained early during the diversification of Actinobacteria as

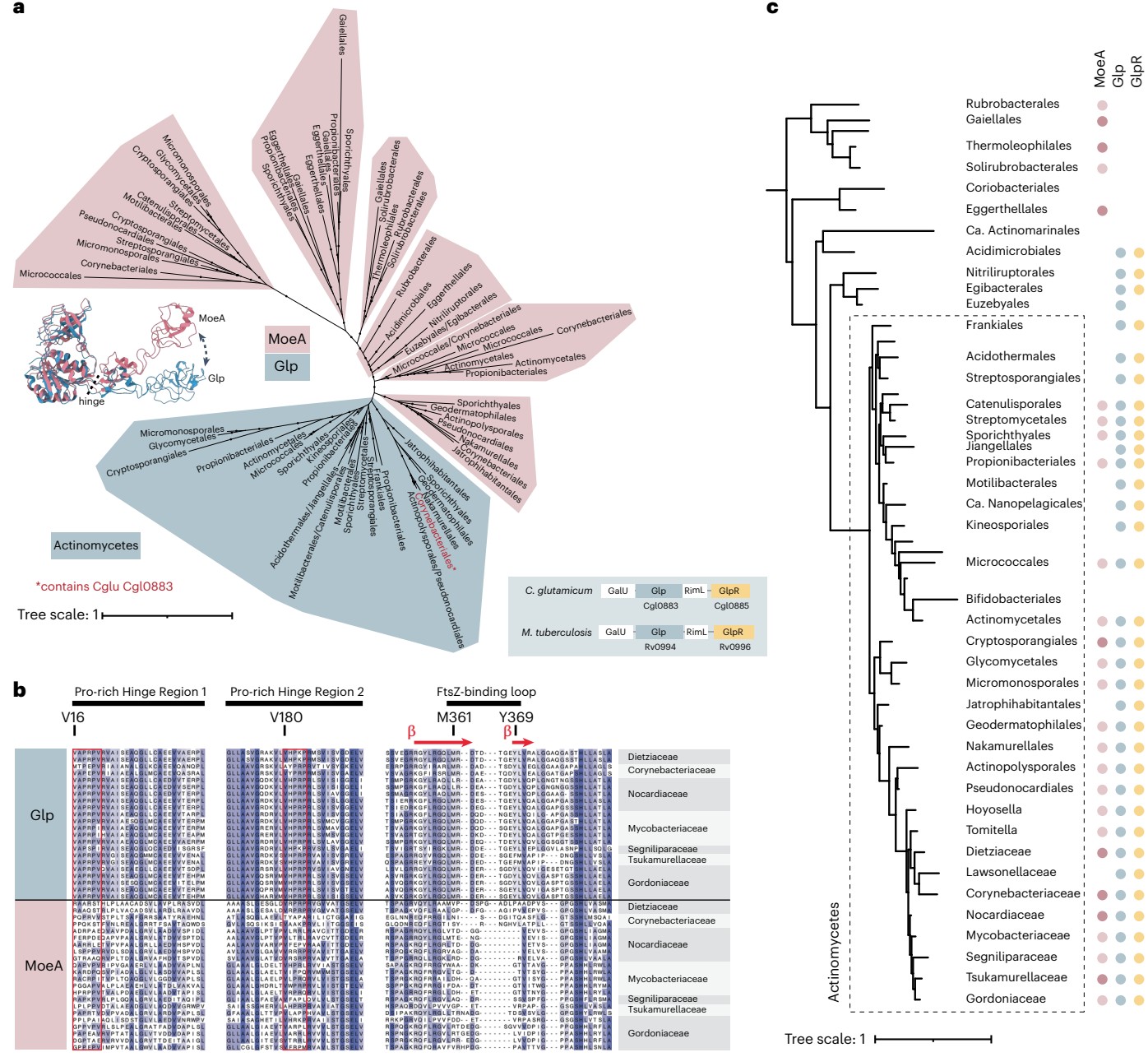

**Fig. 5 | Phylogenetic analyses of Glp in Actinobacteria. a**, Maximum-likelihood phylogeny of MoeA homologues in Actinobacteria. The clade with a blue background corresponds to Glp, clades in pink correspond to other MoeA homologues. Monophyletic classes were collapsed into a single branch for clarity. Dots indicate ultrafast bootstrap (UFB) > 0.85. Scale bar, average number of substitutions per site. For the detailed tree, see Supporting Data. The genomic context of *glp* in *Cglu* and *M. tuberculosis* is indicated on the right of the tree. The locus tags are indicated for genes *glp* and *glpr* present in *Cglu* (Cgl locus tag) and *M. tuberculosis* (Rv locus tag) genomes. **b**, Partial alignment of three selected regions from MoeA paralogues in Corynebacteriales. Sequences of Glp and MoeA are shown for the same species, representative of all Corynebacteriales families. The FtsZ-binding loop is delimited by the key residues methionine (M361) and tyrosine (Y369) indicated according to their position on the *Cglu* sequence. The Pro-rich hinge regions 1 and 2 are indicated by a red rectangle and the first residue inside the box is numbered and highlighted above. **c**, Phyletic pattern for the presence of MoeA, Glp and GlpR in Actinobacteria. Full circles indicate presence of the gene in >50% of the analysed genomes of the phylum. Column MoeA indicates the presence of one (light pink) or more (dark pink) paralogues, except for Glp that is indicated in a separate column. The presence of GlpR is indicated by yellow dots. The phyletic pattern is represented on a reference Actinobacteria tree. Actinobacteria classes were collapsed into a single branch for clarity. Actinomycetes class is indicated by a dashed line. Dots indicate UFB > 0.85. Scale bar, average number of substitutions per site. For the detailed tree, see Extended Data Fig. 7, and for the detailed analysis, see Supplementary Table 3.

a duplication of *moeA* and was inherited vertically by its members. The genomes of most other bacterial phyla contain one *moeA* paralogue that does not branch with actinobacterial *glp* (Extended Data Fig. 6a,b) and likely corresponds to the MoeA enzyme, as in *E. coli*. These results suggest that Glp is a molybdotransferase-related enzyme that has acquired novel functions in Actinobacteria as a result of divergent evolution, to carry out cellular processes specific to the physiology of this order of Bacteria (divisome–elongasome transition).

## Discussion

In this work we have identified Glp, a gephyrin-like repurposed molybdotransferase MoeA enzyme, and its membrane receptor, GlpR, as

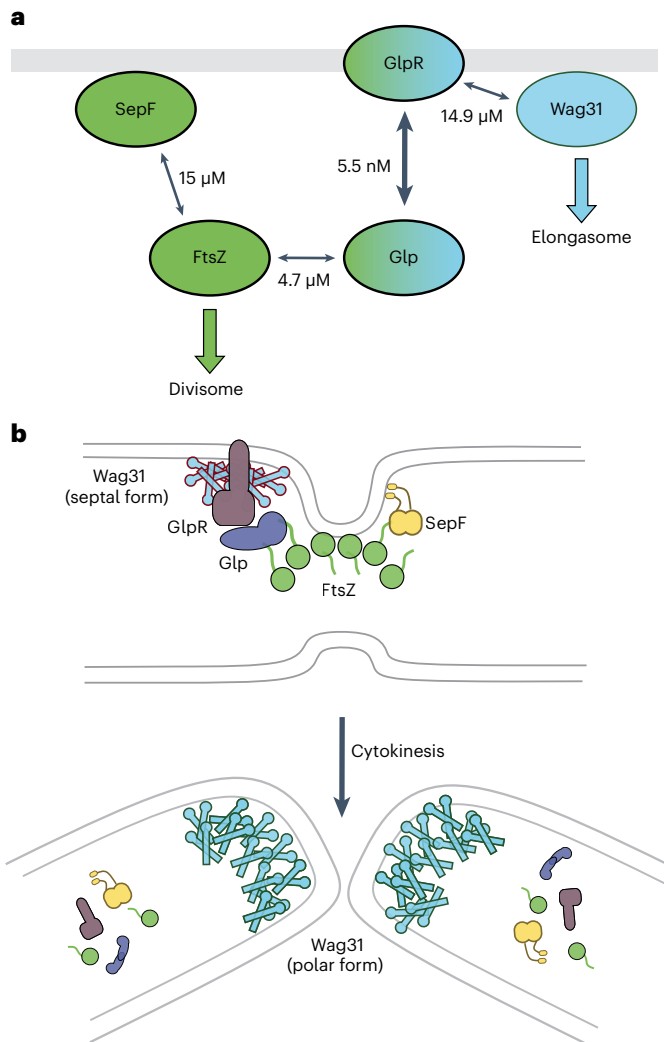

**Fig. 6 | Interaction network and proposed function for Glp–GlpR. a**, Known direct protein–protein interactions and their associated apparent $K_d$ values. Note that SepF-FtsZ was determined using surface plasmon resonance[7], whereas all other measurements were done with BLI (this work). **b**, Working model of the roles of Glp-GlpR-Wag31 in the divisome–elongasome transition during cytokinesis in Corynebacteriales. At the septum, Glp–GlpR would control the functional status of Wag31 and prevent premature pole formation through excessive Wag31 accumulation. Once cell division is completed, this septal control on Wag31 would disappear and an elongation-competent cell pole could form.

that creates the FtsZ-binding pocket for a 2:2 stoichiometric complex. Importantly, these results show that, as in other well-studied bacteria, the corynebacterial FtsZ$_{CTD}$ also acts as a hub for protein–protein interactions in complex and dynamic protein–protein association networks that govern cell division.

Wag31 appears at mid-cell very early in the cell cycle and accumulates asymmetrically at the cell poles over time[22]. This coiled-coil protein has a high propensity to self-associate and build higher-order assemblies or networks[43] (full-length Wag31 easily forms gels in vitro and large foci of Wag31 are seen at the cell poles in vivo). As Wag31 arrives at the septum well before daughter cell separation, its self-association must be controlled to avoid premature pole formation. This negative regulation might depend on conformational states of Wag31, protein concentration and/or post-translational modifications. Indeed, phosphorylation of Wag31 has been shown to be functionally important in *Mycobacteria* as well as *Streptomyces*[17,44]. Whichever the case, the initial control of Wag31 accumulation at mid-cell most likely depends on the divisome. Evidence for this comes from cellular studies where, upon conditional depletion of essential divisome components such as SepF, FtsQ and others[7,45,46], cells start branching (that is, they assemble new poles in erroneous places over the lateral cell walls). Although Wag31 is most abundantly localized at the poles in WT cells, neither Glp nor GlpR are found there (Figs. 1e and 4a). This is consistent with FtsZ retaining Glp/GlpR at the septum to specifically interact with and regulate Wag31, possibly in a coiled-coil conformation differing from the polar one, to preclude premature pole formation. The here identified network could thus make FtsZ a direct regulator of early elongasome assembly and maturation, implying that the Z-ring cytoskeleton would ultimately be responsible for the septal localization of Wag31 (Fig. 6a). Timely removal of this control by Z-ring disassembly and cytokinesis would lead to full maturation of the new pole linked to further Wag31 aggregation and late polar elongasome assembly (Fig. 6b).

We observed erratic pole formation when overexpressing GlpR-mNeon (Fig. 4c), indicating a possible functional interference of the mNeon tag on the FtsZ-Glp-GlpR-Wag31 network and subsequent Wag31 delocalization to induce polar growth in incorrect sites. However, removing Glp or the C-terminal IDR2 from GlpR results in an elongated multiseptal phenotype and septal elongasome dysregulation, but not branching (Figs. 1b,c and 4c–e), probably because these deletions are less disruptive for network formation. These observations suggest that interfering with the regulatory function of Glp/GlpR leads to a delay in cell separation, possibly by dysfunction of the early septal elongasome. Early elongasome assembly should start while the divisome is still in place, as the two protein machineries are thought to be required for the synthesis of the full cell wall at the septum before cytokinesis. Two distinct enzyme systems exist to incorporate the septal and polar peptidoglycan, orchestrated by the SEDS pair of enzymes FtsW/FtsI (divisome) and RodA/b-PBP (elongasome), respectively. However, the synthesis and incorporation of the outer layers of the cell envelope are catalysed by a unique set of enzymes, which belong to the elongasome but are also required to finalize cell separation[11]. Evidence for this comes from experiments done with the anti-tuberculosis drug ethambutol, which targets arabinosyltransferases (EmbA–C) and affects specifically the elongasome[47], but not sPG assembly and divisome function. Although we cannot exclude other effects of Glp/GlpR on cell wall metabolism, their differential sensitivity to ethambutol (Extended Data Fig. 4c) lends some support to the Glp–GlpR complex acting as a bridge between the late divisome and the early elongasome, and makes this system an interesting target for anti-mycobacterial drug development.

In humans, gephyrin is an essential protein for clustering GlyR and GABA receptors at the inhibitory synapse, a process mediated by the underlying tubulin and/or actin cytoskeletons[48–51]. Although the gephyrin–GlyR and Glp–GlpR complexes are involved in unrelated

components of the corynebacterial divisome. We show that Glp and GlpR are central elements of a protein–protein interaction network directly connecting the cytoskeletal proteins from the divisome (FtsZ) and the elongasome (Wag31) (Fig. 6a). The structure of Glp is consistent with its annotation as MoeA, the enzyme involved in the synthesis of the molybdenum co-factor (Moco), which is present in all forms of life and is used by molybdoenzymes to mediate essential cellular functions such as energy generation and detoxification reactions, and to support virulence in pathogenic bacteria[41,42]. MoeA-containing proteins have also acquired additional functions and act as moonlighting proteins[33]. While this feature was thought to be a recent evolutionary trait restricted to eukaryotic Moco biosynthetic enzymes[29], the results presented here show that this is also the case for Glp in Corynebacteriales. The crystal structure of the Glp–FtsZ complex reveals the precise mode of binding of the FtsZ$_{CTD}$ to Glp and provides a molecular signature for the evolutionary repurposing of the molybdotransferase. Glp has evolved a specific grove at the homodimer interface

biological processes, we can draw a molecular analogy between these two networks. They both have undergone evolutionary repurposing from a common enzymatic scaffold (MoeA), and both interact with or organize the tubulin cytoskeleton. They are associated with the membrane through a tight complex formed between MoeA and their membrane receptor, with affinities in the low micromolar or nanomolar range (this work and refs. [52–54]). Until now, repurposing of gephyrin was thought to be a trait reserved to recently evolved species such as *Homo sapiens*[32]. In light of our results on bacterial Glp, it remains an open question whether MoeA repurposing and its link to the tubulin cytoskeleton are inherited traits or evolutionarily independent events. In any case, it appears that the MoeA scaffold has a propensity to acquire functions related to network formation and control at the inner membrane of cells in crucially important processes such as mammalian synaptic signalling and bacterial cell division.

## Methods

### Bacterial strains, plasmids and growth conditions
All strains and plasmids used are listed in Supplementary Table 5. *E. coli* DH5α or CopyCutter EPI400 (Lucigen) were used for cloning and grown in Luria-Bertani (LB) broth or agar plates at 37 °C, supplemented with 50 µg ml$^{-1}$ kanamycin when required. For protein production, *E. coli* BL21 (DE3) was grown in 2YT broth supplemented with auto-induction medium (0.5% glycerol, 0.05% glucose, 0.2% lactose) and 50 µg ml$^{-1}$ kanamycin or 50 µg ml$^{-1}$ carbenicillin. *C. glutamicum* ATCC13032 (*Cglu*) was used as the wild-type strain. *Cglu* strains were grown in brain heart infusion (BHI) or CGXII minimal medium[55] at 30 °C and 120 r.p.m., supplemented with 25 µg ml$^{-1}$ kanamycin when required. For overexpression, CGXII containing 4% sucrose was supplemented with 1% gluconate.

### Ethambutol sensitivity assay
Overnight BHI cultures of *Cglu* and derivative strains were normalized to an optical density (OD)$_{600}$ of 0.5, serially diluted and spotted (10 µl) onto BHI agar medium with and without 1 µg ml$^{-1}$ ethambutol as indicated. Plates were incubated for 24 h at 30 °C and imaged using a ChemiDoc imaging system (BioRad).

### Cgl0883 (glp) and Cgl0885 (glpr) deletion in *C. glutamicum*
We used the two-step recombination strategy with the pk19mobsacB plasmid to delete the coding region of Glp. We amplified approximately 600 bp upstream and downstream of *glp* or *glpr* using chromosomal DNA of *Cglu* as a template. The PCR fragments were cloned by Gibson assembly into a linearized pk19mobsacB, obtaining the plasmid pk19-Δ*glp* or pk19-Δ*glpr*. Plasmids were sequence verified (Eurofins) and electroporated into *Cglu*. Insertion of the plasmids was checked by colony PCR and positive colonies were cultured overnight. The second round of recombination was selected on BHI plates containing 10% (w/v) sucrose. Kanamycin-sensitive colonies were screened by colony PCR for deletions. Positive colonies were sequence verified (Eurofins).

### Cloning for recombinant protein production in *E. coli*
Primers used are listed in Supplementary Table 6. Cloning was performed by assembling the purified PCR fragments into the specified pET derivative expression vector using Gibson assembly with the commercially available NEBuilder HiFi DNA Assembly Cloning kit (New England Biolabs).

The *glp* gene was amplified by PCR using genomic DNA of *Cglu* as template and cloned into a pET vector containing an N-terminal 6xHis-SUMO tag. Glp$_{\Delta Loop}$ (residues 362–371 were replaced by a glycine) was generated by site-directed mutagenesis using the plasmid pET-SUMO-Glp as template.

The gene coding for Wag31 was amplified using gDNA of *Cglu* as template and cloned into a pET vector containing an N-terminal 6xHis-SUMO tag. The plasmid encoding Wag31 with an N-terminal 6xHis tag (pET-His-TEV-Wag31) was synthesized by Genscript. Wag31$_{1–61}$ was generated from the pET-His-TEV-Wag31 plasmid by introducing a STOP codon by PCR mutagenesis after residue 61.

The *glpr* and *glpr*$_{IDR1}$ (residues R24–R214) genes were amplified by PCR using gDNA of *Cglu* as template and cloned into a pET vector containing an N- or C-terminal 6xHis tag. PCR products were digested with DpnI and transformed into chemio-competent *E. coli* cells. All plasmids were verified by Sanger sequencing (Eurofins).

### Cloning for recombinant protein expression in *C. glutamicum*
For recombinant expression in *Cglu*, we used the pUMS_3 shuttle vector, where the gene of interest was placed under the control of P$_{gntK}$, a tight promoter repressed by sucrose and induced by gluconate. Genes were assembled in this plasmid by Gibson or site-directed mutagenesis using the primers listed in Supplementary Table 6. For cellular localization studies, codon-optimized mNeonGreen was ordered from Genscript and cloned alone or fused in frame to the N terminus of *glp*, *moeA1* (*CglO212*) and *moeA3* (*Cgl1196*) constructs, including a GSGS linker between the fused proteins, or at the C terminus of *glpr*. *glpr* and *glpr*$_{\Delta IDR2}$ expression vectors were generated from the pUMS_3-GlpR-mNeon vector by introducing a STOP codon by PCR mutagenesis after *glpr* residues 337 and 266, respectively. Plasmids generated are listed in Supplementary Table 6.

### Protein expression and purification
N-terminal 6xHis-SUMO-tagged Glp and Glp$_{\Delta loop}$ were expressed in *E. coli* BL21 (DE3) following an auto-induction protocol[56]. After 4 h at 37 °C, cells were grown for 20 h at 20 °C in 2YT supplemented with auto-induction medium and 50 µg ml$^{-1}$ kanamycin, collected and flash frozen in liquid nitrogen. Cell pellets were resuspended in 50 ml lysis buffer (50 mM HEPES (pH 8), 500 mM NaCl, 20 mM imidazole, benzonase, EDTA-free protease inhibitor cocktails (ROCHE)) at 4 °C and sonicated. The lysate was centrifuged for 30 min at 30,000 × *g* at 4 °C and loaded onto an Ni-NTA affinity chromatography column (HisTrap FF crude, GE Healthcare). His-tagged proteins were eluted with a linear gradient of buffer B (50 mM HEPES (pH 8), 500 mM NaCl, 500 mM imidazole). The eluted fractions containing the protein of interest were pooled and dialysed in the presence or absence of the SUMO protease (ratio 1:40). Dialysis was carried out at 18 °C overnight in 20 mM HEPES (pH 7.5) and 500 mM NaCl. Cleaved His-tags and His-tagged SUMO protease were removed with Ni-NTA agarose resin. The SUMO-tagged or cleaved proteins were concentrated and loaded onto a Superdex 200 16/60 size exclusion (SEC) column (GE Healthcare) pre-equilibrated at 4 °C in 20 mM HEPES (pH 7.5) and 500 mM NaCl. The peak corresponding to the protein was concentrated, flash frozen in small aliquots in liquid nitrogen and stored at −80 °C.

6xHis-SUMO-FtsZ was produced as described above except for changes in buffer composition: lysis buffer (50 mM HEPES (pH) 8, 300 mM KCl, 5% glycerol, 1 mM MgCl$_2$, benzonase, lysozyme, 0.25 mM TCEP, EDTA-free protease inhibitor cocktails (Roche)), buffer B (50 mM HEPES (pH 8), 300 mM KCl, 5% glycerol, 1 M imidazole). The eluted IMAC fractions containing the protein of interest were pooled and dialysed in the presence or absence of the SUMO protease (ratio used 1:100). Dialysis was carried out at 4 °C overnight in 25 mM HEPES (pH 8), 150 mM KCl and 5% glycerol. Cleaved His-tags and His-tagged SUMO protease were removed with Ni-NTA agarose resin. The protein was concentrated and loaded onto a Superdex 200 16/60 SEC column (GE Healthcare) at 4 °C in 25 mM HEPES (pH 8), 150 mM KCl and 5% glycerol. The pure protein was concentrated, flash frozen in liquid nitrogen and stored at −80 °C.

N- or C-terminal 6xHis-tagged GlpR was expressed in *E. coli* BL21 (DE3) using auto-induction. Cell pellets were resuspended in 150 ml lysis buffer (50 mM HEPES (pH 7.5), 50 mM NaCl, 1 mM MgCl$_2$, benzonase, EDTA-free protease inhibitor cocktails (ROCHE)) at 4 °C and loaded 3 times in a CellD disrupter (Constant Systems). The lysate was

centrifuged for 15 min at 12,000 × $g$ at 4 °C and the supernatant was centrifuged again for 1 h at 100,000 × $g$ at 4 °C. The pellet containing the membrane fraction was resuspended in 40 ml membrane buffer (50 mM HEPES (pH 7.5), 500 mM NaCl, 10 mM imidazole, 10% glycerol, 1% DDM and EDTA-free protease inhibitor cocktails (Roche)) and incubated for 30 min at 4 °C. The membrane solubilized fraction was incubated for 1 h at 4 °C with 1 ml of Ni-resin (Super Ni-NTA resin, Neo Biotech). Beads were collected and washed with 10 column volumes of IMAC A buffer (50 mM HEPES (pH 7.5), 300 mM NaCl, 25 mM imidazole, 10 % glycerol, 0.05% DDM) and His-tagged GlpR was eluted with 10 column volumes of IMAC B buffer (50 mM HEPES (pH 7.5), 300 mM NaCl, 500 mM Imidazole, 10% glycerol, 0.05% DDM). The eluted fractions containing the protein of interest were pooled, concentrated and loaded onto a Superdex 200 10/300 SEC column (GE Healthcare) pre-equilibrated at 4 °C in SEC buffer (25 mM HEPES (pH 7.5), 150 mM NaCl, 10% glycerol, 0.05% DDM). The pure protein was concentrated, flash frozen in small aliquots in liquid nitrogen and stored at −80 °C.

6xHis-TEV-Wag31$_{1-61}$ (DivIVA domain) was expressed in *E. coli* BL21 (DE3) using auto-induction. Cell pellets were resuspended in 50 ml lysis buffer (50 mM HEPES (pH 7), 500 mM NaCl, 10 mM imidazole, 10% glycerol, benzonase, lysozyme, EDTA-free protease inhibitor cocktails (Roche)) at 4 °C and sonicated. The lysate was centrifuged for 1 h at 20,000 × $g$ at 4 °C and loaded onto an Ni-NTA affinity chromatography column (HisTrap FF crude, GE Healthcare). His-tagged protein was eluted with a linear gradient of buffer B (50 mM HEPES (pH 7), 500 mM NaCl, 10% glycerol, 1 M imidazole). The eluted fractions containing the protein of interest were pooled and dialysed with the TEV protease (ratio 1:25). Dialysis was carried out at 18 °C overnight in 50 mM HEPES (pH 7), 150 mM NaCl and 5% glycerol. Cleaved His-tags and His-tagged TEV protease were removed with Ni-NTA agarose resin. The cleaved protein was concentrated and loaded onto a Superdex 200 16/60 SEC column (GE Healthcare) pre-equilibrated at 4 °C in 50 mM HEPES (pH 7), 150 mM NaCl and 5% glycerol. The pure protein was concentrated, flash frozen in liquid nitrogen and stored at −80 °C.

6xHis-SUMO-Wag31 was expressed in *E. coli* BL21 (DE3) using auto-induction. Cell pellets were resuspended in 50 ml lysis buffer (20 mM HEPES (pH 7), 150 mM NaCl, benzonase, EDTA-free protease inhibitor cocktails (Roche)) at 4 °C and lysed by sonication. The lysate was centrifuged for 30 min at 30,000 × $g$ at 4 °C. After centrifugation, a gel-like layer containing His-SUMO-Wag31 was formed between the cell debris pellet and the clarified supernatant. This gel was recovered, washed 3 times with lysis buffer and solubilized in buffer (20 mM HEPES (pH 8.5), NaCl 150 mM). Solubilized SUMO-Wag31 was digested overnight with SUMO protease (ratio used 1:100) at 18 °C. Cleaved His-tags and His-tagged SUMO protease were removed with Ni-NTA agarose resin. Wag31 protein was dialysed overnight at 4 °C in buffer (20 mM HEPES (pH 8.5), 150 mM NaCl), concentrated, flash frozen in liquid nitrogen and stored at −80 °C.

For antibody production, N-terminal 6xHis-SUMO-GlpR$_{IDR1}$ was expressed and purified according to the same protocol as 6xHis-SUMO-tagged Glp described above, except that the final step was carried out on a Superdex 75 16/60 SEC column pre-equilibrated at 4 °C in 20 mM HEPES (pH 8) and 150 mM NaCl. The pure protein was concentrated, flash frozen in liquid nitrogen and stored at −80 °C.

All purified proteins used in this work were run on SDS−PAGE and shown in Supplementary Fig. 1.

## Crystallization

Crystallization screens were done using the sitting-drop vapour diffusion method and a Mosquito nanolitre-dispensing crystallization robot at 18 °C (TTP Labtech)[57]. Optimal crystals of Glp (13.5 mg ml$^{-1}$) were obtained in 100 mM Tris (pH 8.5), 30% (v/v) PEG400 and 200 mM Na$_3$Cit. For single-wavelength anomalous diffraction (SAD) phasing, Glp crystals were soaked in mother liquor containing 10 mM Cl$_4$K$_2$Pt for 30 min. The complex of Glp bound to the FtsZ$_{CTD}$ peptide (DDLDVPSFLQ, purchased from Genosphere) was crystallized at 0.34 mM Glp (15 mg ml$^{-1}$) and 1.7 mM FtsZ$_{CTD}$. Crystals appeared after 2 weeks in 0.1 M NaCl, 0.1 M Bis-Tris (pH 6.5) and 1.5 M (NH4)$_2$SO$_4$. Crystals were cryo-protected in mother liquor containing 33% (v/v) ethylene glycol or 33% (v/v) glycerol.

## Data collection, structure determination and refinement

X-ray diffraction data were collected at 100 K using synchrotron radiation (Supplementary Table 2) at Soleil (France) and processed using XDS[58] and AIMLESS from the CCP4 suite[59]. A 2.35 Å dataset from a Glp crystal soaked in Cl$_4$K$_2$Pt was used to solve the structure by SAD phasing using Phaser[60] and automatic model building with Buccaneer both from the CCP4 suite. The structures of Glp alone and in complex with FtsZ$_{CTD}$ were refined through iterative cycles of manual model building with COOT[61] and reciprocal space refinement with Phenix[62]. The final crystallographic statistics and the PDB deposition codes of the atomic coordinates and structure factors are shown in Supplementary Table 2. Structural figures were generated with ChimeraX[63].

## Differential scanning fluorescence (thermofluor) assay

Glp (3 μg) in 25 mM HEPES (pH 8), 150 mM NaCl and 5% glycerol with or without 1 mM FtsZ$_{CTD}$ was dispensed into 96-well PCR plates (20 μl per well in triplicates). 50X Sypro Orange (0.6 μl, Invitrogen) was added and the mixture heated from 25 to 95 °C in steps of 1 °C per min in a CFX96 Touch Real-Time PCR Detection system (BioRad). Excitation and emission filters (492 and 516 nm, respectively) were used to monitor the fluorescence increase. The midpoint of the protein unfolding transition was defined as the melting temperature $T_m$.

## BLI assays

Experiments were performed on the Octet-Red384 device (Pall Forte-Bio) at 25 °C. To test interactions between FtsZ and Glp variants, the His-SUMO_Glp variants were diluted at 227 nM in capture buffer (20 mM HEPES (pH 7.5), 500 mM NaCl, 1 mg ml$^{-1}$ BSA) and immobilized on Sartorius Ni-NTA biosensors for 10 min at 1,000 r.p.m., followed by a washing step of 2 min. Empty sensors were used as reference for non-specific binding. FtsZ was diluted at 10 μM in polymerization buffer (25 mM Pipes (pH 6.9), 100 mM KCl, 10 mM MgCl$_2$) and pre-incubated with or without 1 mM guanosine-5'-triphosphate at room temperature for 20 min. Binding of FtsZ to the immobilized His-SUMO_Glp variants was monitored for 10 min with agitation at 1,000 r.p.m., followed by dissociation in the same buffer without proteins for 10 min. In the reciprocal approach, His-SUMO-FtsZ was diluted at 4 μM in polymerization buffer supplemented with 1 mg ml$^{-1}$ BSA and then immobilized on the Ni-NTA biosensors for 10 min at 1,000 r.p.m., followed by a washing step in the same buffer. Sensors loaded with His-LysA (an unrelated protein) were used as control for non-specific binding. His-SUMO-FtsZ-loaded or reference sensors were incubated for 3 min at 1,000 r.p.m. in the absence or presence of 2-fold serially diluted concentrations of Glp (80–1.25 μM range) in polymerization buffer, followed by dissociation in the same buffer without protein for another 3 min. To test the interaction between Glp, GlpΔ$_{loop}$ and GlpR, C-terminal His-tagged GlpR was immobilized at the surface of Ni-NTA biosensors and untagged Glp or GlpΔ$_{loop}$ were tested for binding to GlpR. Empty sensors were used as reference. GlpR was diluted at 0.4 μM in 25 mM HEPES (pH 7.5), 150 mM NaCl, 10% glycerol, 0.05% DDM and 1 mg ml$^{-1}$ BSA, and then immobilized on Ni-NTA biosensors. Association of untagged Glp variants in GlpR buffer was monitored for 30 min, followed by dissociation in the same buffer without proteins for another 30 min.

For the biotinylation reaction, 100 μl of His-GlpR at 25 μM was incubated with 20x molar excess of EZ-Link NHS-PEG4-Biotin (Thermo Scientific) following supplier instructions. Biotinylated GlpR was diluted to 0.25 μM in GlpR buffer (25 mM HEPES (pH 7.5) 150 mM NaCl, 10% glycerol, 0.05% DDM) and immobilized on the commercially available Sartorius Streptavidin biosensors for 5 min at 1,000 r.p.m., followed by

a washing step in GlpR buffer. Empty sensors were used as reference. GlpR-loaded or empty reference sensors were incubated for 5 min at 1,000 r.p.m. in the absence or presence of 2-fold serially diluted concentrations of Wag31 (150–2.34 µM range) in buffer A (20 mM HEPES (pH 8.5), 150 mM NaCl, DDM 0.05%) or DivIVA (200–3.15 µM range) in buffer B (25 mM HEPES (pH 7.5), 150 mM NaCl, 10% glycerol, 0.05% DDM, 1 mg ml$^{-1}$ BSA). Specific signals were obtained by double referencing, subtracting non-specific signals measured on reference sensors and buffer signals on specific loaded sensors. Assays were performed at least twice. To obtain the $K_d$ values, steady-state signal versus concentration curves were fitted using GraphPad Prism 9, assuming a one-site binding model.

## Circular dichroism

All measurements were acquired with an Aviv 215 spectropolarimeter. Far-UV (195–260 nm) spectra were recorded at 25 °C using a 0.2-mm-path-length cylindrical cell. GlpΔ$_{loop}$ was measured at 20 µM in a buffer containing 20 mM HEPES (pH 7.5) and 500 mM NaCl. Ellipticity was measured every 0.5 nm and averaged over 2 s. The final spectrum was obtained by averaging 3 successive scans and subtracting the baseline spectrum of the buffer recorded under the same conditions. BestSel[64] was used for quantitative decomposition of the far-UV circular dichroism spectrum.

## Co-IP of Wag31/GlpR in *C. glutamicum*

*Cglu*, *Cglu_Δglp* or *Cglu_Δglpr* strains were grown in CGXII minimal medium at 30 °C for 6 h. Cells were collected, washed with 1X PBS and normalized by resuspending cell pellets in PBS-T (1X PBS, 0.1% (v/v) Tween-80) to a final OD$_{600}$ of 10. The cell suspensions were cross-linked with formaldehyde (0.25% v/v) for 20 min at 30 °C with gentle agitation. The crosslinking reaction was stopped by adding 1.25 M glycine and incubated for 5 min at room temperature. Sample preparation and co-IP was performed as described above with magnetic agarose beads coupled to anti-GlpR antibodies. Eluted samples were subjected to immunoblot analysis using anti-GlpR or anti-DivIVA antibodies.

## Cell fractionation

*Cglu strains* were grown in BHI medium at 30 °C for 6 h and collected by centrifugation. Bacterial cell pellets were resuspended in lysis buffer (25 mM Tris (pH 7.5), 150 mM NaCl, benzonase, EDTA-free protease inhibitor cocktails (Roche)) and disrupted at 4 °C with 0.1 mm glass beads and using a PRECELLYS 24 homogenizer. Cell debris and aggregated proteins were removed by centrifugation at 14,000 × g for 20 min at 4 °C. The supernatant was recentrifuged at 90,000 × g for 30 min at 4 °C to pellet cell membranes. Membrane fractions were solubilized with lysis buffer + 0.5% SDS. Protein concentrations were determined using UV$_{280}$ absorbance and adjusted to 6 mg ml$^{-1}$. Of each fraction, 120 µg was run on an SDS–PAGE gel and analysed by western blot using anti-Glp antibodies.

## Antibody production, purification and characterization

Anti-Glp, anti-GlpR and anti-Wag31 antibodies were raised in rabbits (Covalab) against purified Glp, GlpR$_{IDR1}$ or Wag31$_{1-61}$ antigens. For antibody purification, sera from day 67 post inoculation were purified using a 1 ml HiTrap NHS-activated HP column (GE Healthcare) loaded with the corresponding antigen according to manufacturer instructions. Sera were diluted in binding buffer (20 mM sodium phosphate (pH 7.4), 500 mM NaCl), loaded onto the column and washed with 7 ml of binding buffer. Antibodies were eluted with 10 ml elution buffer (100 mM glycine (pH 3), 500 mM NaCl) and neutralized with 1 M Tris (pH 9). Purified antibodies were concentrated to 8 mg ml$^{-1}$ and mixed 1:1 with 100% glycerol, aliquoted and stored at −20 °C. The characterization of the antibodies is shown in Supplementary Fig. 3. Anti-SepF and anti-mScarlet antibody production was described previously[7].

## Western blots

For cell extracts, pellets were resuspended in lysis buffer (50 mM Bis-Tris (pH 7.4), 75 mM 6-aminocaproic acid, 1 mM MgSO4, benzonase and protease inhibitor) and disrupted at 4 °C with 0.1 mm glass beads and using a PRECELLYS 24 homogenizer. Total extracts (6–120 µg) were run on an SDS–PAGE gel, transferred onto a 0.2 µm nitrocellulose membrane and incubated for 1 h with blocking buffer (5% skimmed milk, 1X TBS-Tween buffer) at r.t. Blocked membranes were incubated for 1 h at r.t., with the corresponding primary antibody diluted to the appropriate concentration in blocking buffer. After washing in TBS-Tween buffer, membranes were probed with an anti-rabbit or an anti-mouse horseradish peroxidase-linked secondary antibody (GE healthcare) for 45 min. For chemiluminescence detection, membranes were washed with 1X TBS-T and revealed with HRP substrate (Immobilon Forte, Millipore). Images were acquired using the ChemiDoc MP imaging system (BioRad). All uncropped blots are shown in Supplementary Fig. 3. Dilutions used: anti-Glp (1:500), anti-GlpR (1:500), anti-Wag31 (1:500), anti-mNeonGreen (1:1,000), anti-mouse and anti-rabbit secondary Abs (1:10,000).

## Mass spectrometry

**Sample preparation for mass spectrometry analysis.** For SepF interactome, *Cglu* and the strains expressing SepF-Scarlet, SepF$_{K125E/F131A}$-Scarlet and Scarlet were grown in CGXII minimal media supplemented with 1% gluconate for 6 h at 30 °C. Cell suspensions were cross-linked with 0.25% (v/v) formaldehyde for 20 min at 30 °C and protein extracts were incubated with magnetic beads with 10 µg of purified antibodies covalently linked[7] (anti-Scarlet or anti-SepF produced by Covalab). Proteins recovered after washing and elution with 1 M glycine (pH 2) were neutralized with 1 M Tris (pH 9), denatured (2 M urea), reduced (10 mM DTT, 1 h, r.t.), alkylated (55 mM IAM, 45 min, r.t.) and digested with 0.5 µg of trypsin (Promega). Tryptic peptides were desalted using a POROS R2 resin (Thermo Fisher), vacuum dried and resuspended in 0.1% formic acid (FA). GLP interactomes were obtained in two strains (*Cglu* and *Cglu_Δglpr*) using anti-GLP antibodies and the experimental conditions described above, except for the protein reduction, alkylation and tryptic digestion steps that were performed using the FASP protocol[65] with filter passivation in 5% Tween-20 (ref. 66). *Cglu_Δglp* was used as control. For all interactomes, tryptic peptides from 4 biological replicates for each condition were analysed using a nano-HPLC–MS/MS. To calculate protein enrichment, we analysed the full proteome of *Cglu* grown in CGXII minimal media from 1 cm long SDS–PAGE gels (12% acrylamide) using 3 biological replicates. In-gel reduction, Cys alkylation and digestion were performed in the conditions described above.

## Nano-HPLC–MS/MS

Tryptic peptides were analysed using a nano-HPLC (UltiMate 3000, Thermo) coupled to a Q-Exactive Plus mass spectrometer (Thermo). Peptide mixtures were loaded onto a precolumn (Acclaim PepMapTM 100, C18, 75 µm × 2 cm) and separated on a C$_{18}$ Easy-Spray column (PepMapTM RSLC, 75 µm × 50 cm) using a two-solvent system: (A) 0.1% FA in water and (B) 0.1% FA in acetonitrile at a flow rate of 200 nl min$^{-1}$. The separation gradients used were: from 0% to 55% B in 65 min for the SepF interactome; from 1% to 35% B over 90 min for the GLP interactome and over 150 min for the total proteome.

MS analysis was carried out in data-dependent mode (MS followed by MS/MS of the top 12 ions) using dynamic exclusion. The survey scans were acquired from 200 to 2,000 $m/z$ with a resolution of 70,000 at 200 $m/z$, while MS/MS spectra were acquired at a resolution of 17,500 at 200 $m/z$.

## Protein identification and data analysis

PatternLab for Proteomics V (PatternLabV) was used to perform peptide spectrum matching and label-free quantitation analyses on the

basis of extracted-ion chromatograms (XIC)[67]. XICs were obtained by integrating the intensity of a given peptide's ion current over a narrow mass-to-charge window as a function of retention time. For the SepF interactome, a target reverse database was generated using the Cglu ATCC13032 proteome downloaded from UniProt (November 2018), to which the sequences of Scarlet, SepF-Scarlet, SepF$_{K125E/F131A}$-Scarlet and the most common contaminants in proteomics experiments were added. For the GLP interactome and the global proteome, a target reverse Cglu ATCC13032 database downloaded from UniProt (November 2021) including the most common contaminants in proteomics was used. Search parameters were set as follows: $m/z$ precursor tolerance: 35 ppm, methionine oxidation and cysteine carbamidomethylation as variable and fixed modifications, respectively, and a maximum of two missed cleavages. Search results were filtered to achieve a false discovery rate value of ≤1% at protein level and 10 ppm tolerance for precursor ions.

To identify SepF interactors, we compared the list of proteins recovered under different conditions: WT strain using α-SepF antibodies (WT/α-SepF), SepF-Scarlet strain using α-SepF antibodies (SepF-Scarlet/α-SepF) and SepF-Scarlet strain using α-Scarlet antibodies (SepF-Scarlet/α-Scarlet). As a control for background binding, the Scarlet strain using α-Scarlet antibodies was used. In addition, we compared proteins recovered from SepF-Scarlet and SepF$_{K125E/F131A}$-Scarlet using α-Scarlet antibodies. To identify GLP interactors, we compared the proteins recovered from *Cglu* or *Cglu_Δglpr* with *Cglu_Δglp* strain.

PatternLab's Venn diagram statistical module was used to determine proteins uniquely detected in each biological condition using $P < 0.05$ (ref. 68). In addition, pairwise comparison between Co-IPs and controls was performed using the XIC browser on the basis of the Benjamini–Hochberg theoretical estimator to deal with multiple $t$-tests and the following conditions: maximum parsimony, minimum number of peptides of 1, minimum number of MS1 counts of 5 and $\log_2$FC > 1.8.

Enrichment factors for SepF interactors were calculated as the ratio of the normalized spectral abundance factor of each interactor in the interactome to that in the proteome. To compare the recovery of SepF interactors in pull-down analyses of SepF-Scarlet and Sep-F$_{K125E/F131A}$-Scarlet strains, the signal (ΣXIC of detected peptides in each replicate) was normalized by the signal of SepF in the corresponding sample. Statistical analysis was performed using a two-sided unpaired Student's $t$-test ($P < 0.05$). All data are presented as mean ± s.d. Calculations were done using GraphPad Prism. The same approach was used to compare the recovery of Wag31 in the GLP interactome of *Cglu* and *Cglu_Δglpr* strains.

## Phase contrast and fluorescence microscopy

For imaging, cultures were grown in BHI for ~6 h, pelleted at 5,200 × $g$ at r.t. and inoculated into CGXII, 4% sucrose and kanamycin (25 µg ml$^{-1}$) for overnight growth. The following day, cultures were diluted to an OD$_{600}$ of 1 in CGXII and 4% sucrose (±1% gluconate), and grown for 6 h to an OD$_{600}$ of ~5 (early exponential phase). Cultures (100 µl) were pelleted, washed with fresh medium and diluted to an OD$_{600}$ of 3 for imaging. For membrane staining, Nile Red (2 µg ml$^{-1}$ final, Enzo Life Sciences) was added to the culture before placing them on 2% agarose pads prepared with corresponding growth media. Cells were visualized using a Zeiss Axio Observer Z1 microscope fitted with an Orca Flash 4 V2 sCMOS camera (Hamamatsu) and a Pln-Apo ×63/1.4 oil Ph3 objective. Images were collected using Zen Blue 2.6 (Zeiss) and analysed using the software Fiji[69] and the plugin MicrobeJ[70] to generate violin plots and fluorescence intensity heat maps. Heat maps represent the averaged localization of the mNeon-tagged protein on a representative cell. For all analyses, the *Cglu* strain corresponds to *Cglu* + empty plasmid. For *Cglu_Δglpr* + GlpR-mNeon, only cells showing a mean intensity of mNeon fluorescence >35,000 were considered, to discard cells that lost the plasmid.

## Statistics and reproducibility

Because of the important number of cells analysed in each sample, Cohen's $d$ value was used to describe effect sizes between different strains independently of sample size:

$$d = \frac{\text{mean}_2 - \text{mean}_1}{\sqrt{\frac{(n_1-1)\text{s.d.}_1{}^2+(n_2-1)\text{s.d.}_2{}^2}{n_1+n_2-2}}}$$

Values were interpreted according to the reference intervals suggested by Cohen[71] and expanded by Sawilowsky[72] as follows: small (ns), $d < 0.50$; medium (*), $0.50 < d < 0.80$; large (**), $0.80 < d < 1.20$; very large (***), $1.20 < d < 2.0$; huge (****), $d > 2.0$.

Unless otherwise stated, $P$ values were obtained using a Welch two-sample $t$-test calculated in R. All experiments were performed as biological triplicates. Some autofluorescence was observed for wild-type *Cglu* as previously described[7]. All micrographs and blots shown are representative of similar experiments carried out at least three times, except for those corresponding to Figs. 3c and 4g, and Extended Data Fig. 1d that were performed only once.

## Protein database assemblies

For the sequence analyses in Actinobacteria, we assembled two databases representing all Actinobacteria diversity present at the National Center for Biotechnology (NCBI) as of February 2021: ACTINO_DB (244 taxa) and ACTINO_REDUCED_DB (113 taxa from ACTINO_DB). For ACTINO_DB, we selected 5 species per class, except for class Corynebacteriales, where we selected 5 species per order (Supplementary Table 3). For the phylogenetic analyses in Bacteria, we assembled a database on the basis of the one provided in ref. 73. We reduced the taxonomic sampling to 76 species by removing all candidate phyla (Supplementary Table 4).

## Homology searches and mapping

To identify all MoeA homologues in the ACTINO_DB, we used HMM profile searches. We used the HMMER package (v.3.3.2)[74] tool 'jackhmmer' to look for homologues of Glp and GlpR in all the proteomes, respectively using the GenBank[75] sequences BAB98276.1 and BAB98278.1 as query. The hits were aligned with mafft (v.7.475)[76] using default parameters. Alignments were manually curated, removing sequences that did not align. The hits obtained by jackhmmer might not include sequences that are very divergent from the single sequence query. For this, the curated alignments were used to create HMM profiles using the HMMER package tool 'hmmbuild'. Curated HMM profiles for Glp and GlpR were used for a final round of searches against the ACTINO_DB and Bacteria databases, using the HMMER tool 'hmmsearch'. All hits were curated to remove false positives by checking alignments obtained using linsi, the accurate option of mafft (v.7.475). For Actinobacteria, hits were also curated on the basis of their genomic context. We retrieved 5 genes upstream and downstream of each MoeA paralogue, identified and grouped the corresponding proteins into protein families. Each family larger than 10 sequences was used to create HMM profiles as explained before. These profiles, together with the Glp and GlpR profiles, were used in MacSyFinder[77] against the ACTINO_DB to identify conserved genomic contexts containing MoeA and 3 or more members of these families separated by no more than 5 other proteins and a permissive $e$-value (<0.1). This analysis complemented the GlpR homology searches, as the sequences are very divergent and therefore difficult to identify without their genomic context. Finally, we analysed the taxonomic distribution of the identified Glp and GlpR sequences. The phyletic pattern and the genomic context information were mapped on an Actinobacteria reference phylogeny using iTOL[78] and custom scripts.

## Phylogenetic analyses

The alignments of Glp homologues (including all MoeA paralogues) were trimmed using BMGE (v.1.2)[79] (option -m BLOSUM30) to keep

only informative positions. These alignments were used to reconstruct the phylogeny of the MoeA paralogues in ACTINO_REDUCED_DB and Bacteria. We used the maximum-likelihood phylogeny reconstruction tool IQ-TREE (v.2.0.6)[80], with the LG + F + R8 and LG + F + R10 models (-m MFP), respectively, and ultrafast bootstraps (-B 1000).

To reconstruct the reference phylogeny of ACTINO_DB, we concatenated the protein sequences of RNApol subunits B, B' and IF-2. Homologues of these proteins were identified, aligned and trimmed as explained before. These alignments were concatenated into a supermatrix to infer a maximum-likelihood tree with IQ-TREE, using the posterior mean site frequency (PMSF) and the model LG + C60 + F + I + G, with ultrafast bootstrap supports calculated from 1,000 replicates. The guide tree required by the PMSF model was obtained using the MFP option and the same supermatrix. The reference phylogeny of Bacteria was obtained from ref. 73 and candidate taxa were pruned from the tree.

### Reporting summary

Further information on research design is available in the Nature Portfolio Reporting Summary linked to this article.

## Data availability

Atomic coordinates and structure factors have been deposited in the PDB with accession codes 8BVE (Glp) and 8BVF (Glp–FtsZ$_{CTD}$). The mass spectrometry proteomics data have been deposited to the ProteomeXchange Consortium via the PRIDE[81] partner repository with the dataset identifier PXD037255. All phylogenetic data used to produce our results are provided as Supporting Data at https://doi.org/10.17632/265wyk8r3f.1. All materials of this paper can be provided upon reasonable request. Custom scripts will be made available upon request. Source data for all relevant figures are provided with this paper.

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

## Acknowledgements

We thank the core facilities at the Institut Pasteur C2RT, P. England, B. Raynal, S. Brûlé (PFBMI); P. Weber, C. Pissis, A. Mechaly (PFC), and J. Fernandes and A. Salles (UtechS PBI / Imagopole, supported by France BioImaging; ANR-10–INSB–04; Investments for the Future); P. Campagne from the Bioinformatics and Biostatistics Hub from the IP-C3BI; the synchrotron source Soleil (Saint-Aubin, France) for granting access to the facility and the staff of Proxima 1 and Proxima 2A beamlines for helpful assistance during X-ray data collection. Molecular graphics were done with ChimeraX, developed at UCSF with support from NIH (R01-GM129325) and NIAID. This work was supported in part by grants from the Agence Nationale de la Recherche (ANR, France), contracts ANR-18-CE11-0017 (P.M.A.) and ANR-21-CE11-0003 (A.M.W.); Agencia Nacional de Investigacion et Innovacion (ANII, Uruguay), FCE_1_2019_1_155569 (R.D.), FOCEM-COF 03/11 (R.D.), ECOS-Sud France-Uruguay, contract U20B02 (A.M.W. and R.D.); and by institutional grants from the Institut Pasteur, the CNRS, and Université Paris Cité. J.P. was funded through the AMX programme from the Ecole Polytechnique. A.S. was part of the Pasteur–Paris University (PPU) International PhD Program, funded by the European Union's Horizon 2020 research and innovation programme under Marie Sklodowska-Curie grant agreement no. 665807. Q.G. was funded by MTCI PhD school (ED 563). A.R. was funded by ANII, CAP (UdelaR) and PEDECIBA.

## Author contributions

A.M.W., R.D. and P.M.A. designed the research. M.M., J.P., A.L., A.S., Q.G., M.B.A. and A.M.W. conducted the protein biochemistry, cell biology and genetic experiments, and purified proteins for structural and biophysical studies. J.P., A.D., C.G. and A.M.W. performed cellular imaging and analysis. M.M. and J.P. carried out the biochemical and biophysical studies of protein–protein interactions. A.L., M.M., A.R., M.M.P., A.M.W. and R.D. carried out MS and interactomics experiments. M.M., A.S., A.H. and P.M.A. carried out the crystallogenesis and crystallographic studies. D.M. performed the phylogeny and sequence analyses. A.M.W. and P.M.A. wrote the paper. All authors edited the paper.

## Competing interests

The authors declare no competing interests.

## Additional information

**Extended data** is available for this paper at https://doi.org/10.1038/s41564-023-01473-0.

**Correspondence and requests for materials** should be addressed to Rosario Durán or Anne Marie Wehenkel.

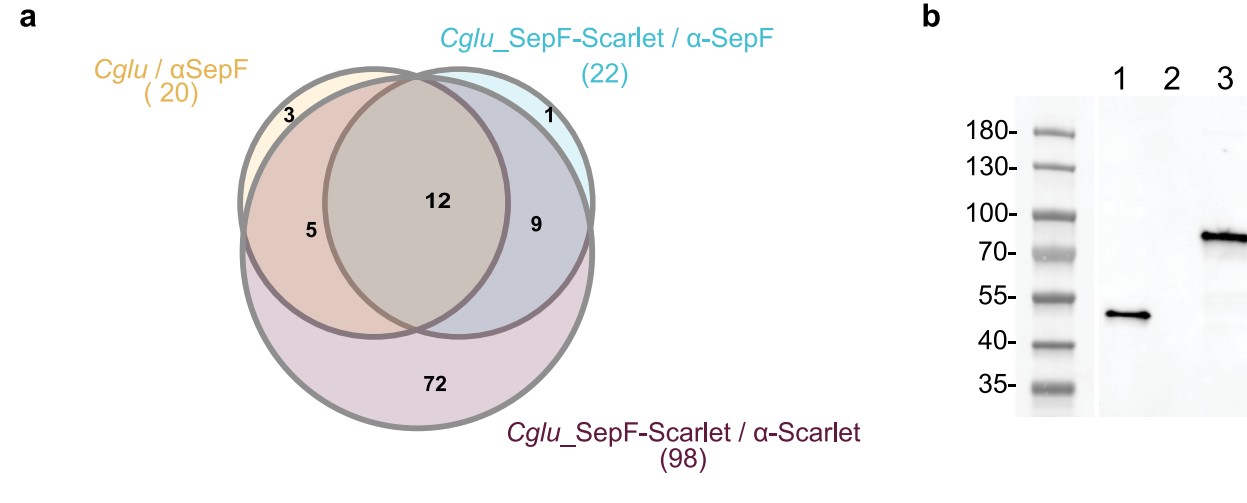

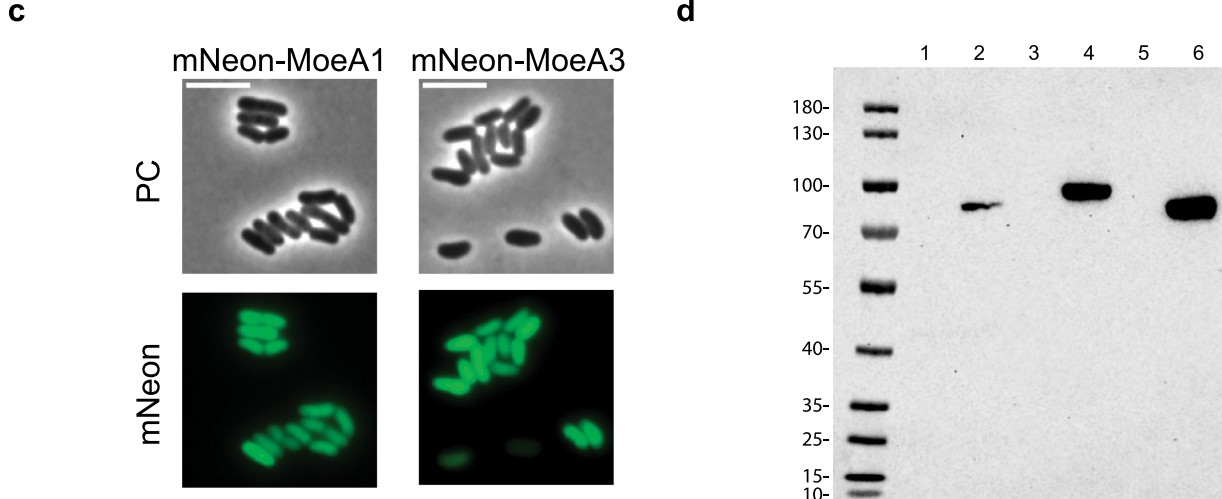

**Extended Data Fig. 1 | Identification of Glp as a cell division protein. (a)** Venn diagram showing the overlap between 3 independent SepF interactomes using *Cglu* or *Cglu*_SepF-Scarlet strains. Proteins only detected in each interactome were identified by comparison with control condition and using the probability mode (*p* value < 0.05) of Patternlab Venn diagram module following a bayesian model[68]. Proteins enriched in SepF Co-IPs when compared to controls were identified using pairwise comparison module of Patternlab V based on XIC intensities. 20, 22 and 98 proteins were detected as SepF interactors in *Cglu*/α-SepF (strain/antibody), *Cglu*_SepF-Scalet/α-SepF and *Cglu*_SepF-Scalet/α-Scarlet respectively. 12 proteins were common to all of Co-IPs, and for 11 of them an enrichment factor in relation to the total proteome could be calculated, and thus represent the core SepF interactome (Supplementary Table 1,a and Fig. 1a). One additional interactor, the hypothetical protein Cgl1805, could not be detected in the proteome and no enrichment factor could thus be reported. **(b)** Western blots of whole cell extracts (120 μg) from *Cglu* (lane 1) and *Cglu_Δglp* strains

transformed with the empty plasmid (lane 2) or mNeon-Glp (lane 3). Glp and mNeon-Glp levels were revealed using the α-Glp antibody. Left: molecular weight markers (kDa) **(c)** Cellular localization of *Cglu* MoeA homologs MoeA1/*Cgl0212* (25% aa sequence identity) and MoeA3/*Cgl1196* (27% aa sequence identity). Representative images in phase contrast and mNeon fluorescent signal for *Cglu*_mNeon-MoeA1 and *Cglu*_mNeon-MoeA3. Both MoeA1 and MoeA3 are cytosolic, which contrasts with the mid-cell localization of mNeon-Glp shown in Fig. 1e. All Scale bars 5μm. **(d)** the cytoplasmic distribution was not due to fusion protein degradation as shown by Western blots of whole cell extracts (120 μg) from *Cglu* carrying mNeon-MoeA1, mNeon-Glp or mNeon-MoeA3 plasmids and revealed using an α-mNeon antibody. Left: molecular weight markers (kDa); Lanes 1: mNeon-MoeA1 (sucrose); 2: mNeon-MoeA1 (gluconate); 3: mNeon-Glp (sucrose); 4: mNeon-Glp (gluconate); 5: mNeon-MoeA3 (sucrose); 6: mNeon-MoeA3 (gluconate).

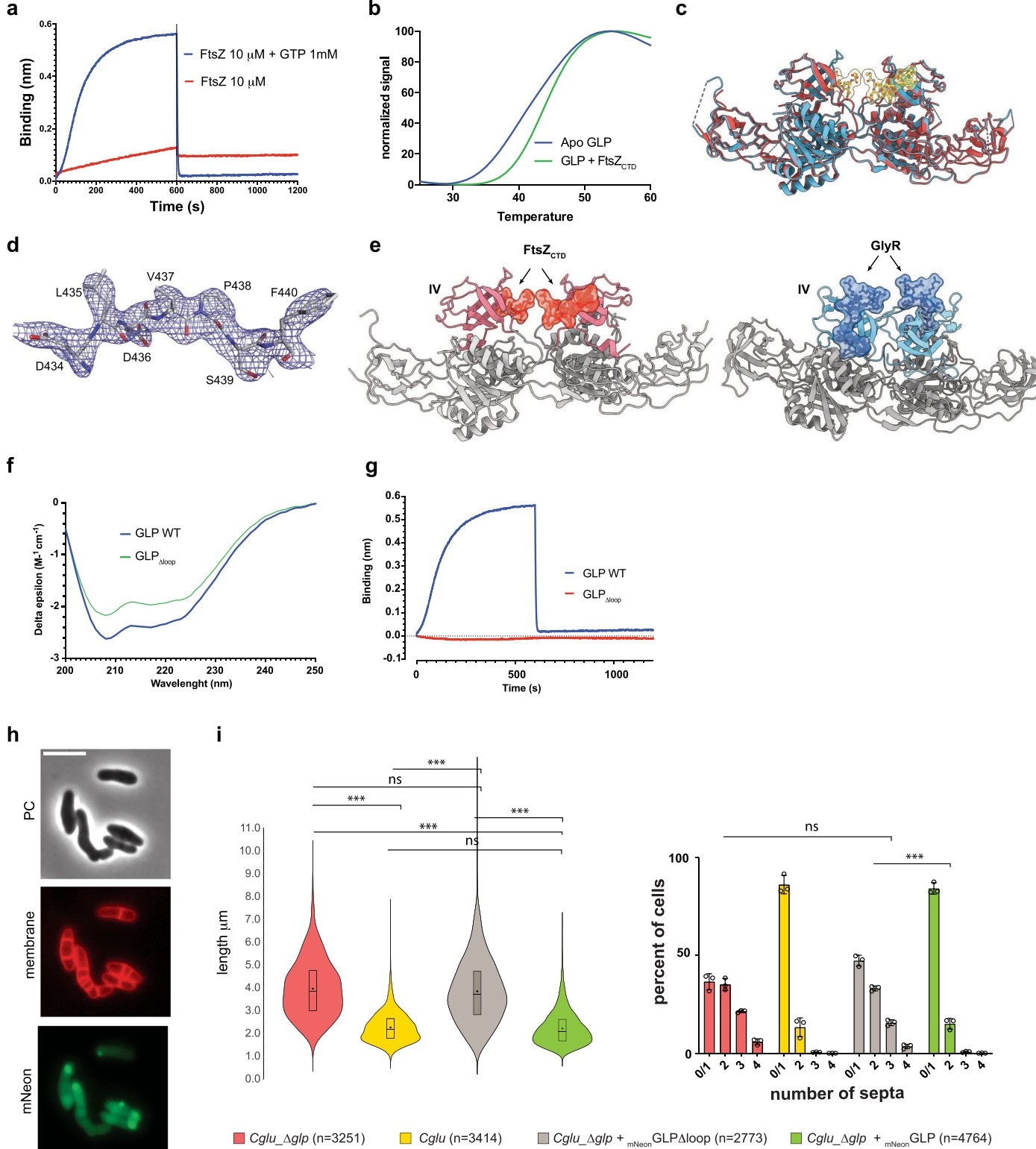

**Extended Data Fig. 2 | See next page for caption.**

**Extended Data Fig. 2 | Glp-FtsZ interaction.** (**a**) BLI sensorgrams of FtsZ binding to immobilized Glp in the presence or absence of 1 mM GTP. (**b**) Normalized melting curves of Glp with or without 1 mM FtsZ$_{CTD}$ peptide as determined by a thermofluor assay. Glp was stabilized by 2 °C in the presence of FtsZ$_{CTD}$. (**c**) The crystal structures of ligand-free (red) and FtsZ$_{CTD}$-bound (blue and yellow ligand). Glp can be superimposed with an r.m.s.d. of 0.64 Å for 392 equivalent Cα atoms. (**d**) Electron density map of FtsZ$_{CTD}$ bound to Glp monomer B contoured at 1.2 σ. (**e**) Comparison of the Glp-FtsZ$_{CTD}$ complex (red, left panel) with the Gephyrin-GlyR complex (blue, right panel, PDB code: 2fts). In both cases the FtsZ$_{CTD}$ and GlyR peptides (molecular surfaces) bind the domain IV (in colour) of Glp and gephyrin respectively. (**f**) Far-UV circular dichroism spectra of Glp and Glp$_{\Delta loop}$. (**g**) BLI sensorgrams of FtsZ (10 μM) binding to immobilized Glp or Glp$_{\Delta loop}$. (**h**) Representative images for mNeon-Glp$_{\Delta loop}$ in *Cglu_Δglp*. Scale bar = 5 μm.

(**i**) Left, Violin plots showing the distribution of cell length. The number of cells used in the analyses (n) is indicated below each plot representing triplicate experiments. The box indicates the 25$^{th}$ to the 75$^{th}$ percentile, the mean and the median are indicated with a dot and a line in the box, respectively. Significance indicated corresponds to values of Cohen's d (from top to bottom: (***, d = 1,55, P = 0), (ns, d = 0,09, P = 0,0012), (***, d = 1,76, P = 0), (***, d = 1,58, P = 0), (***, d = 1,78, P = 0), (ns, d = 0, P = 0,95)). Right, frequency histogram indicating the number of septa per cell, calculated from n cells imaged from 3 independent experiments (triplicates) for each strain (for *Cglu*, n = 718, 1468 and 1223; for *Cglu_Δglp* + mNeon-Glp, n = 1641, 1311 and 1801; for *Cglu_Δglp*, n = 873, 1538 and 840; for *Cglu_Δglp* + mNeon-Glp$_{\Delta loop}$, n = 678, 1187 and 905); open circles represent the corresponding data points; bars represent the mean ± SD. Cohen's d from top to bottom (ns, d = 0.28, P = 1.19e-26), (***, d = 1.25, P = 0).

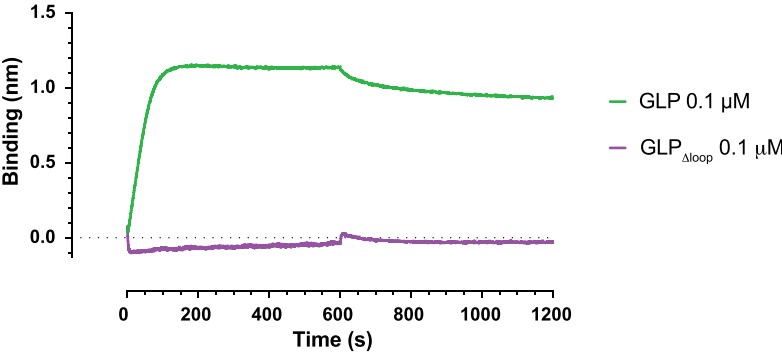

**Extended Data Fig. 3 | BLI sensorgram.** BLI sensorgram of Glp or Glp$_{\Delta loop}$ (0.1 µM) binding to immobilized GlpR.

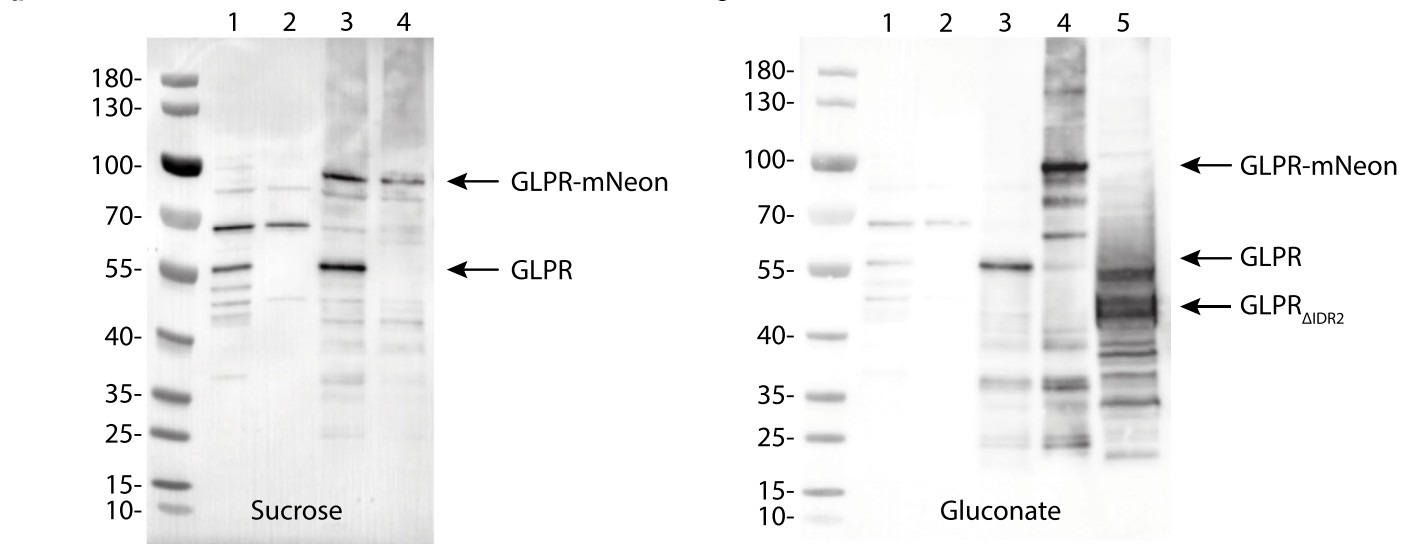

**d**

Western Blot corresponding to 4a

**e**

Western Blot corresponding to 4c

**Extended Data Fig. 4 | See next page for caption.**

**Extended Data Fig. 4 | Phenotypic analysis of the *Cglu_Δglpr* strain.**
(**a**) Western blots of whole cell extracts (120 μg) from *Cglu* and *Cglu_Δglpr*. GlpR was revealed using an α-GlpR antibody. An arrow indicates the specific signal for GlpR. Molecular weight markers (kDa) are shown on the left (**b**) Representative images of *Cglu_Δglpr*. Scale bar 5μm. Violin plots showing the distribution of cell length (ns, d = 0,46, p = 8,22e-75) and cell width (ns, d = 0, p = 0,29) for *Cglu_Δglpr* and *Cglu*. The number of cells (n) used (from triplicates) is indicated below each violin representation; the box indicates the 25$^{th}$ to the 75$^{th}$ percentile, mean and the median are indicated with a dot and a line in the box, respectively. Frequency histogram indicating the number of septa per cell for *Cglu* and *Cglu_Δglpr* strains, calculated from n cells imaged from 3 independent experiments for each strain (for *Cglu*, n = 718, 1468 and 1223; for *Cglu_Δglpr*, n = 931, 934 and 1363); open circles represent the corresponding data points; bars represent the mean ± SD. Cohen's d (ns, d = 0.01, p-value = 0.8271459) (**c**) Ethambutol sensitivity assay of Δ*glpr strain*. (**d**) Western blots of whole cell extracts (120 μg) from *Cglu* and *Cglu_Δglpr* strains complemented with the empty plasmid or GlpR-mNeon. GlpR was revealed using an α-GlpR antibody. An arrow indicates the specific signal for GlpR and GlpR-mNeon. Western blot corresponds to representative cells shown in Fig. 4a. Left: molecular weight markers (kDa); Lane 1: *Cglu* + empty plasmid; Lane 2: *Cglu_Δglpr* + empty plasmid; Lane 3: *Cglu* + GlpR-mNeon; Lane 4: *Cglu_Δglpr* + GlpR-mNeon. (**e**) Western blots of whole cell extracts (120 μg) from *Cglu* and *Cglu_Δglpr* strains complemented with the empty plasmid, GlpR, GlpR-mNeon or GlpR$_{ΔIDR2}$. GlpR was revealed using an α-GlpR antibody. Arrows indicate specific signal for GlpR, GlpR-mNeon and GlpR$_{ΔIDR2}$. Western blot corresponds to representative cells shown in Fig. 4c. Left: molecular weight markers (kDa); Lane 1: *Cglu* + empty plasmid; Lane 2: *Cglu_Δglpr* + empty plasmid; Lane 3: *Cglu_Δglpr* + GlpR; Lane 4: *Cglu_Δglpr* + GlpR-mNeon; Lane 5: *Cglu_Δglpr* + GlpR$_{ΔIDR2}$. Note that many cells in the GlpR-mNeon overexpressing strain have lost their plasmid and thus the overexpression is underestimated in these whole cell extracts.

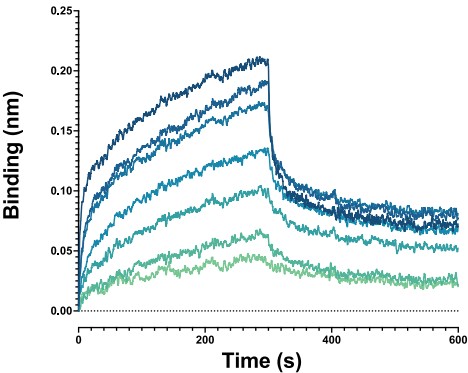

**Extended Data Fig. 5 | GlpR-Wag31 interactions.** Sensorgrams of Wag31$_{1-61}$ binding to immobilized GlpR by biolayer interferometry. A series of measurements using a range of concentrations for Wag31$_{1-61}$ (200 μM (dark blue) - 3.125 μM (light green)) was carried out to derive the apparent equilibrium dissociation constant $Kd$ (14.86 μM) from steady-state signal versus concentration curves fitted assuming a one-site binding model.

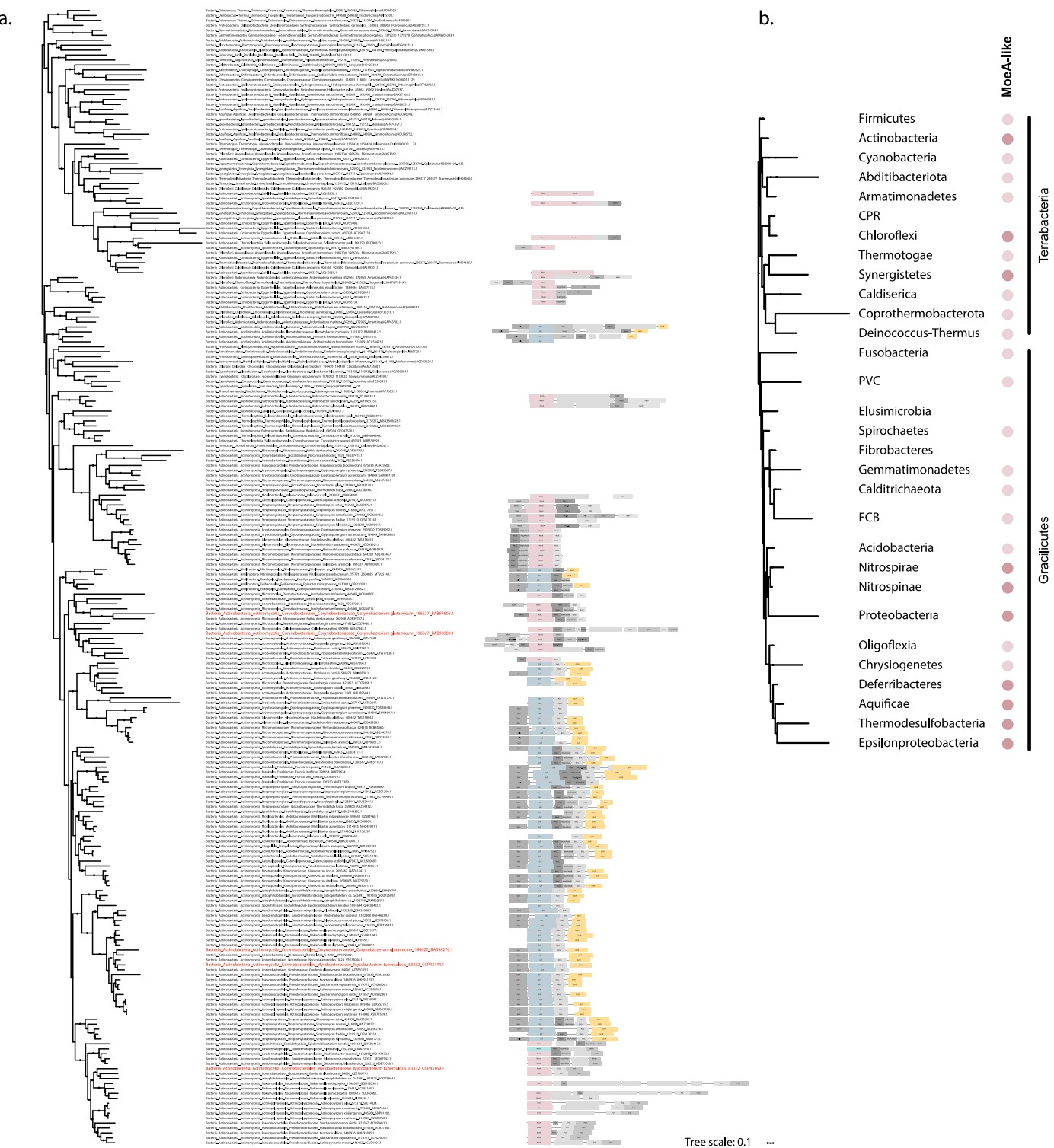

**Extended Data Fig. 6 | Phylogenetic analyses of Glp in Bacteria. (a)** Maximum likelihood phylogeny of MoeA-like paralogs in Bacteria. The genomic context of Glp/MoeA paralogs is indicated for each branch, if conserved in at least two other cases. Non-conserved genes in the genomic context are indicated with gray rectangles without labels. Branches that correspond to *C. glutamicum* and *M. tuberculosis* species are indicated in red. Dots indicate UFB > 0.85.

The scale bar represents the average number of substitutions per site. **(b)** Phyletic pattern for the presence of MoeA-like paralogs in Bacteria. Full circles indicate presence of the gene in more than 50% of the analyzed genomes of the phylum, darker pink indicates the presence of more than one copy. The phyletic pattern is represented on a reference Bacteria tree[73]. Phyla were collapsed into a single branch for clarity. For the detailed analysis see Supplementary Table 4.

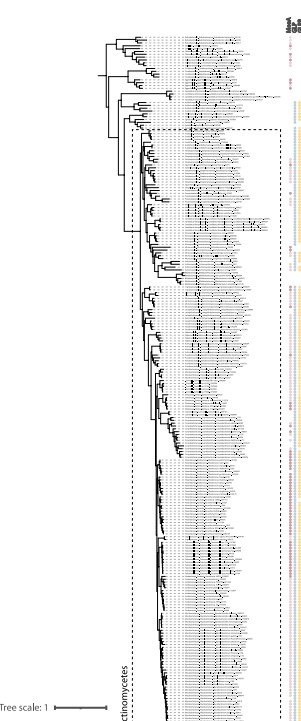

**Extended Data Fig. 7 | Phyletic pattern for the presence of MoeA, Glp and GlpR in *Actinobacteria*.** Extended version of Fig. 5c. Full circles indicate presence of the gene in the species and blanks indicate its absence. In the column MoeA, darker pink indicates the presence of more than one copy. Column MoeA represents all paralogs, except for Glp that is indicated in a separate column. The phyletic pattern is represented on a reference *Actinobacteria* tree. *Actinomycetes* class is indicated by a dashed line. Dots indicate UFB > 0.85. The scale bar represents the average number of substitutions per site.

Anne Marie Wehenkel

# Reporting Summary

## Statistics

For all statistical analyses, confirm that the following items are present in the figure legend, table legend, main text, or Methods section.

| n/a | Confirmed | |
|---|---|---|
| ☐ | ☒ | The exact sample size (*n*) for each experimental group/condition, given as a discrete number and unit of measurement |
| ☐ | ☒ | A statement on whether measurements were taken from distinct samples or whether the same sample was measured repeatedly |
| ☐ | ☒ | The statistical test(s) used AND whether they are one- or two-sided<br>*Only common tests should be described solely by name; describe more complex techniques in the Methods section.* |
| ☐ | ☒ | A description of all covariates tested |
| ☐ | ☒ | A description of any assumptions or corrections, such as tests of normality and adjustment for multiple comparisons |
| ☐ | ☒ | A full description of the statistical parameters including central tendency (e.g. means) or other basic estimates (e.g. regression coefficient) AND variation (e.g. standard deviation) or associated estimates of uncertainty (e.g. confidence intervals) |
| ☐ | ☒ | For null hypothesis testing, the test statistic (e.g. *F*, *t*, *r*) with confidence intervals, effect sizes, degrees of freedom and *P* value noted<br>*Give P values as exact values whenever suitable.* |
| ☒ | ☐ | For Bayesian analysis, information on the choice of priors and Markov chain Monte Carlo settings |
| ☒ | ☐ | For hierarchical and complex designs, identification of the appropriate level for tests and full reporting of outcomes |
| ☐ | ☒ | Estimates of effect sizes (e.g. Cohen's *d*, Pearson's *r*), indicating how they were calculated |

*Our web collection on statistics for biologists contains articles on many of the points above.*

## Software and code

Policy information about availability of computer code

| Data collection | Microscopy: Zen Blue 2.6 (Zeiss); Soleil synchrotron (site specific data collection software), Biolayer Interferometry (Octet-Red384 V11.1.1.19), Circular dichroism (Aviv 215 software v3.16); Mass spectrometry: nano-HPLC (UltiMate 3000, Thermo) coupled to a hybrid quadrupole-orbitrap mass spectrometer (QExactive Plus, Thermo); Western Blot imaging: ChemiDoc MP Image Lab Touch 3.0.1 (Biorad). |
|---|---|
| Data analysis | Microscopy (Fiji v2.9.0/1.53t; MicrobeJ v5.130); X-ray crystallography (XDS v 20220110, CCP4 suite v8.0, Phenix v1.20.1, Phaser v2.8.2, Coot v09.8.8, ChimeraX v1.5); Biolayer Interferometry (GraphPad Prism 9); Circular dichroism: Bestsel; Mass spectrometry: Pattern Lab for Proteomics V software; Phylogeny: HMMER package (v3.3.2) jackhammer, mafft (v7.475), MacSyFinder v1.0.5, iTOL v6, BMGE (v1.2). |

For manuscripts utilizing custom algorithms or software that are central to the research but not yet described in published literature, software must be made available to editors and reviewers. We strongly encourage code deposition in a community repository (e.g. GitHub). See the Nature Portfolio guidelines for submitting code & software for further information.

## Data

Policy information about availability of data

All manuscripts must include a data availability statement. This statement should provide the following information, where applicable:

- Accession codes, unique identifiers, or web links for publicly available datasets
- A description of any restrictions on data availability
- For clinical datasets or third party data, please ensure that the statement adheres to our policy

Atomic coordinates and structure factors have been deposited in the PDB with accession codes 8BVE (Glp) and 8BVF (Glp-FtsZCTD). The mass spectrometry proteomics data have been deposited to the ProteomeXchange Consortium via the PRIDE81 partner repository with the dataset identifier PXD037255 (http://www.ebi.ac.uk/pride/archive/projects/PXD037255). All phylogenetic data used to produce our results is provided as Supporting Data under the following link: https://data.mendeley.com/datasets/265wyk8r3f/draft?a=6d0ecd8f-0adb-4f71-b984-37b7710c3f0a . All materials of this paper can be provided upon reasonable request. Custom scripts will be made available upon request. Source data are provided for all relevant Figures.

## Human research participants

Policy information about studies involving human research participants and Sex and Gender in Research.

| Reporting on sex and gender | Not applicable |
|---|---|
| Population characteristics | Nor applicable |
| Recruitment | Not applicable |
| Ethics oversight | Not applicable |

Note that full information on the approval of the study protocol must also be provided in the manuscript.

# Field-specific reporting

Please select the one below that is the best fit for your research. If you are not sure, read the appropriate sections before making your selection.

☒ Life sciences      ☐ Behavioural & social sciences      ☐ Ecological, evolutionary & environmental sciences

For a reference copy of the document with all sections, see nature.com/documents/nr-reporting-summary-flat.pdf

# Life sciences study design

All studies must disclose on these points even when the disclosure is negative.

| Sample size | For microscopy images no specific sample size was determined, and in general large numbers of cells were analyzed in order to get close to a normal distribution. Other sample sizes were selected based on published research in the field and/or preliminary experimentation. No sample size calculation was performed. Sample sizes for each experiment are described in detail in Figure legends or Methods section. |
|---|---|
| Data exclusions | For microscopy image analysis: clusters of cells impossible to segment were exluded from analysis, but they were checked manually to verify that no specific phenotypes were excluded. In the analysis corresponding to Figure 4c only cells showing a mean intensity of mNeon fluorescence greater than 35000 were considered, to discard cells that lost the plasmid, due to toxicity of this plasmid. This is stated in the Methods section. No other data were excluded. |
| Replication | For cellular studies, at least 3 independent replicates were used. Cells were grown in equivalent conditions and harvested at similar optical densities to assure exponential growth. For mass spectrometry at least 3 replicates per condition were used. Details are provided in figure legends and methods section (Statistics and reproducibility). |
| Randomization | Samples were not randomized as this is not applicable to this study. |
| Blinding | All samples were collected and analyzed with attributed numbers to avoid any bias in collection or analysis, but further blinding is not necessary for our experiments, as they only involve rational data. |

# Reporting for specific materials, systems and methods

We require information from authors about some types of materials, experimental systems and methods used in many studies. Here, indicate whether each material, system or method listed is relevant to your study. If you are not sure if a list item applies to your research, read the appropriate section before selecting a response.

## Materials & experimental systems

| n/a | Involved in the study |
|---|---|
| ☐ | ☒ Antibodies |
| ☒ | ☐ Eukaryotic cell lines |
| ☒ | ☐ Palaeontology and archaeology |
| ☒ | ☐ Animals and other organisms |
| ☒ | ☐ Clinical data |
| ☒ | ☐ Dual use research of concern |

## Methods

| n/a | Involved in the study |
|---|---|
| ☒ | ☐ ChIP-seq |
| ☒ | ☐ Flow cytometry |
| ☒ | ☐ MRI-based neuroimaging |

## Antibodies

**Antibodies used**

Anti-GLP (polyclonal, Rabbit, Covalab, custom produced)
Anti-GLPR (polyclonal, Rabbit, Covalab, custom produced)
Anti-DivIVA (polyclonal, Rabbit,Covalab, custom produced)
Anti-mNeonGreen (monoclonal, mouse, ChromoTek, Ref: 32F6)
Anti-Mouse (ECL Mouse IgG, HRP-linked whole Ab sheep, Cytiva, Ref: NA931V)
Anti-Rabbit (ECL Rabbit IgG, HRP-linked whole Ab donkey, Cytiva, Ref: NA934V)
Anti-SepF (polyclonal, Rabbit, Covalab custom produced and described previously (Sogues et al, 2020, Nat Comm)
Anti-mScarlet (polyclonal, Rabbit, Covalab custom produced and described previously (Sogues et al, 2020, Nat Comm)

**Validation**

Antibodies were validated against the recombinant antigen used for production and when possible in wild-type versus depleted strains of C. glutamicum. This is described in the Methods and Supplementary information of this work for anti-Glp, anti-GlpR and anti-Wag31 and for anti-SepF and anti-mScarlet in Sogues et al, 2020, Nat Comm. All other antibodies are commercially available and validation statements are available on manufacturers website.

