## [Peer Review File · Nature Microbiology]

Peer Review Information

Journal: Nature Microbiology

Manuscript Title: Eukaryotic-like gephyrin and cognate membrane receptor coordinate corynebacterial cell division and polar elongation.

Corresponding author name(s): Dr Anne Marie Wehenkel, Dr Rosario Durán

Editorial Notes:

This manuscript has been previously reviewed at another journal. This document only contains reviewer comments, rebuttal and decision letters for versions considered at Nature Microbiology. Mentions of prior referee reports have been redacted.

Reviewer Comments & Decisions:Decision Letter, initial version:

Message: 7th March 2023

Dear Dr Wehenkel,

Thank you for your patience while your manuscript "Eukaryotic-like gephyrin and cognate membrane receptor coordinate corynebacterial cell division and polar elongation." was under peer-review at Nature Microbiology. It has now been seen by 3 referees, whose expertise and comments you will find at the end of this email. Although they find your work of some potential interest, they have raised a number of concerns that will need to be addressed before we can consider publication of the work in Nature Microbiology.

All of the reviewers raise numerous technical points that should be addressed with additional analyses or explanation in the text. In particular, Reviewer #3 was concerned that the model was only tenuously supported by the data at present--this should be addressed by the addition of experiments to better support the claims, and/or by adequately caveating this section.

Should further experimental data allow you to address these criticisms, we would be happy to look at a revised manuscript.

Please include a data availability statement as a separate section after Methods but before references, under the heading "Data Availability". This section should inform readers about the availability of the data used to support the conclusions of your study. This information includes accession codes to public repositories (data banks for protein, DNA or RNA sequences, microarray, proteomics data etc...), references to source data published alongside the paper, unique identifiers such as URLs to data repository entries, or data set DOIs, and any other statement about data availability. At a minimum, you should include the following statement: "The data that support the findings of this study are available from the corresponding author upon request", mentioning any restrictions on availability. If DOIs are provided, we also strongly encourage including these in the Reference list (authors, title, publisher (repository name), identifier, year). For more guidance on how to write this section please see:

<http://www.nature.com/authors/policies/data/data-availability-statements-data-citations.pdf>

2If revising your manuscript:

* If you have not done so already we suggest that you begin to revise your manuscript so that it conforms to our Article format instructions at <http://www.nature.com/nmicrobiol/info/final-submission>. Refer also to any guidelines provided in this letter.

[redacted]

Note: This url links to your confidential homepage and associated information about manuscripts you may have submitted or be reviewing for us. If you wish to forward this e-mail to co-authors, please delete this link to your homepage first.

Nature Microbiology is committed to improving transparency in authorship. As part of our efforts in this direction, we are now requesting that all authors identified as 'corresponding author' on published papers create and link their Open Researcher and Contributor Identifier (ORCID) with their account on the Manuscript Tracking System (MTS), prior to acceptance. This applies to primary research papers only. ORCID helps the scientific community achieve unambiguous attribution of all scholarly contributions. You can create and link your ORCID from the home page of the MTS by clicking on 'Modify my Springer Nature account'. For more information please visit www.springernature.com/orcid.

If you wish to submit a suitably revised manuscript we would hope to receive it within 6 months. If you cannot send it within this time, please let us know. We will be happy to consider your revision, even if a similar study has been accepted for publication at Nature

3Microbiology or published elsewhere (up to a maximum of 6 months).

[redacted]

Reviewer Expertise:

Referee #1: cell division, Corynebacteriales, cell wall biology

Referee #2: structural biology, bacterial cell biology

Referee #3: cell division, cell wall biology, genetics

Reviewer Comments:

Reviewer #1 (Remarks to the Author):

The present article presents a comprehensive analysis of the coordination of cell division and polar elongation in Corynebacteria, a group of gram-positive bacteria that are known to cause various types of infections. The authors have identified two new members of the divisome, a complex of proteins that are essential for the proper segregation of chromosomes during cell division, in these organisms. The discovery of these genes is a significant achievement, as it represents the first time that such genes have been studied in Corynebacteria. This research could have far-reaching implications in the development of novel anti-infective agents against related pathogens, as understanding the molecular mechanisms of cell division and elongation in these organisms could lead to the identification of new drug targets.

One of the key findings of the study is that a mutant strain lacking either GLP or GLPR, two of the newly identified divisome members, are viable. This suggests that these genes may not be essential for cell division or cell elongation, and could potentially have redundant functions. Additionally, this implies that other proteins may also be involved in this process. The authors have excluded two other MoeA-like proteins, but it is important to investigate if there are any other candidate proteins in the immunoproteomics data that could have a similar role.

Another noteworthy finding is that a delta GLP mutant can continue to elongate. This has major implications in the analysis of the data, as it suggests that the lack of these coordinators does not have a major effect on elongation or cell division. This, in turn, suggests that the role of GLP/GLPR in these processes is secondary and could be covered to some extent by other proteins. The authors should therefore provide more detailed analysis of this finding and the possible mechanisms underlying this observation.

Finally, the authors should provide growth curves to compare the effect of deleting GLP or GLPR in the replication of *Corynebacterium glutamicum*, as this would provide a more comprehensive evaluation of the impact of these genes on the divisome and elongasome of Corynebacteria. Overall, the study presents a valuable contribution to the field of microbiology, and it is important for the authors to address the aforementioned points in order to further strengthen their findings.

Reviewer #2 (Remarks to the Author):

In this paper, Martinez et al. identified two proteins involved in cell division in

4Corynebacteriales – GLP and GLPR that bear resemblance to eukaryotic gephyrin-like protein and its receptor respectively. Using biochemical and structural approaches, the authors show that GLP binds to the C-terminal of the major bacterial cell division protein, FtsZ and bridges the interaction between the septal cell division machinery and the polar cell elongation machinery via interaction of GLPR with Wag31, the major component of the polar elangosome in Corynebacteriales. It is a well-written paper for a broad audience and shows an important advance in our understanding of the less-common lab bacteria belonging to Corynebacteriales, especially how the activities of components of the septal divisome and the polar elangosome are coordinated in space and time. Parts of the paper which need more explanation and clarity in my opinion are their data on interaction of SepF/GLP which I have mentioned in my report below.

1. Line 58 in Introduction – the authors mention that so far SepF is the only direct interactor of FtsZ in actinobacteria. Recently, Ramos-Leon et al. in 2021 also found SepH in Streptomyces and Mycobacteria that interacts with and regulates FtsZ dynamics. Perhaps, worth including the information as well to keep the readers upto date with the literature.
2. The paper is already very well and clearly written for a broad audience. This is only a recommendation, so feel free to take it or not. I feel in the introduction, it may be useful to explain a little bit more about the polar growth in actinobacteria vs lateral growth in more commonly studied bacteria like E. coli or B. subtilis, even if it is just a few sentences of a primer. This will be very useful for microbiologists who do not work with actinobacteria and help them appreciate as to why the cell division mechanisms are starkly different in actinobacteria compared to other commonly studied bacteria. The authors do mention about the polar growth a little bit but explaining and differentiating it from lateral growth will help the readers appreciate the significance of their findings even better, in my opinion.
3. Any particular reason why the authors didn't use co-IP using FtsZ as a bait directly and used SepF instead?
4. Fig 1b – for the graph showing the no. of septa, it will be nice to have a legend within the figure itself for the Cglu and Cglu_ Δglp strain. Also, may be make the y-axis “percent of cells”
5. Fig 1e and 5a – it is not exactly clear what the heatmap represents. Is the intensity corresponding to the no of cells with that particular localization on the cell? Will be great to explain in methods or text somewhere.
6. As per the results in Fig 1b, Cglu_ Δglp cells, in addition to being longer due to incomplete division, are also wider and the authors notice excessive accumulation of Wag31 in Cglu_ Δglp cells in Fig 5g. Can the authors speculate on why the cells become wider? Is there any evidence linking dysregulation of polar elangosome assembly in Cglu_ Δglp cells with perhaps Wag31 or other elangosome components participating in lateral cell wall build up or is it due to accumulation of lateral cell wall enzymes?
7. Line 169 - It is kind of surprising that the authors could not detect direct binding between SepF and GLP which contradicts their Fig 1a interactomics data. Why do the authors think they could not detect direct interaction between SepF and GLP in vitro – anything to do with the conditions they used? What about in the presence of FtsZ? Can SepF and GLP interact in vivo in the presence of FtsZ? Did the authors consider doing a two-hybrid assay for the same?
8. Fig 2a – what do these normalized XIC intensity values mean? If we look at the absolute nos, the XIC intensity in case of FtsZ/SepF mutant is like SepF WT/GLP interaction. Does it mean that the SepF WT/GLP interaction is weak to begin with?
9. Did the authors look at the localization of mNeon-GLP in SepF mutant? It will be interesting to see that and will further bolster the claim about interaction of GLP with FtsZ.
10. Based on their data and model, is SepF even then necessary for the interaction of FtsZ and GLP? I am kind of left wondering about the chronology of steps for recruitment of cell

division proteins. Does FtsZ recruit SepF first and then GLP or are they recruited independently to FtsZ C-terminal domain? Even in their model in Fig 6b/6a, there doesn't seem to be any interaction b/w SepF and GLP/GLPR complex as per their schematics, then how come GLP was detected in the interactome of SepF? Am I missing any method details or other justification here?

11. Minor point – line 368 – the reference is in a different format from the rest of the article.

Reviewer #3 (Remarks to the Author):

The work by Petit et al. investigates the structure and regulation of the actinobacterial divisome. This is an important topic, as it is quite clear that this phylum of bacteria, which includes major human pathogens and organisms important for biotechnology, grows and divides using mechanisms that are different from well-studied model organisms. The authors use an impressive combination of genetics, biochemistry, structural biology, and cell biology to propose the presence of a novel protein complex that regulates the machinery necessary for septation and elongation. While interesting and clear conclusions can be drawn from this work, a number of important aspects of model are supported by tenuous evidence, making the overall model quite speculative. The major claims of the manuscript are summarized below:

- 1) A previously uncharacterized protein, named Glp, is a divisome component that is important for septation in *C. glutamicum*. This claim is well supported by genetic and biochemical evidence.
- 2) Glp is paralogous to MoaA, indicating that this protein has evolved a new function in actinobacteria that is unrelated to its annotated role in MoCo synthesis. This is interesting and well supported by phylogenetic studies (though the presentation of this work could be greatly improved).
- 3) Glp forms a complex with the co-operonic protein, GlpR. This is supported by pulldowns (after crosslinking) and in vitro biochemical assays. The importance of this interaction in vivo could be further supported by using existing reagents to investigate whether Glp and GlpR depend on each other for septal localization.
- 4) The Glp/GlpR complex regulates septation and/or elongation. This conclusion is supported by reasonably strong, though indirect evidence. A role for GlpR in controlling cellular morphology is demonstrated upon overexpression of a Neon-tagged version of the protein or a mutant protein lacking the GlpR binding site, but deletion of the gene or expression of the native protein has no effect. It's understandable that some redundancy may obscure a phenotype in the knockout, and the evidence obtained by overexpression of the binding site mutant is reasonably compelling.
- 5) GlpR regulates cell elongation via binding to Wag31. While the biochemical data supporting this interaction is adequate, no convincing evidence supports a function for this interaction in the cell. The authors claim that Wag31 "accumulates" at the septum in Δ glp cells, but there is no control to show normal levels of this protein at the septum. Further, they suggest that observing Wag31 at sites of branching in the GlpR-Neon overexpression strain is evidence for functional interplay between the proteins, but Wag31 will always be found at sites of septation/branching. As a result, both this claim and the model presented in Figure 6 are very speculative.

A valuable manuscript could be crafted based on only the first 4 points, concluding that Glp/GlpR contribute to septation in some important but still unclear manner. If the authors wish to make the additional broad mechanistic claims related to Wag31, a significant amount of work would be necessary and it's unclear what the outcome would

be. Either way, the manuscript is not written or organized in a manner that is sufficient for publication, and the following points should be addressed:

- 1) Throughout the manuscript, the supplementary data section is used inappropriately, separating essential experimental controls from the main text data. In general, the reader should not be forced to refer to the supplement to interpret main text figures. Specifically, this organization makes it impossible to evaluate the data, as it is unclear if the experimental and control data are from the same experiment (e.g. complemented strains, etc.).
- 2) The phylogenetic data in figure 1 is not explained adequately. Furthermore, it seems to rely on synteny between Glp and GlpR, but the latter is not yet introduced. I highly recommend moving this figure and discussion to the end of the manuscript, and explaining the reasoning more thoroughly.
- 3) The BLI studies seem sound, but they are presented like they were copied from a lab notebook. These panels should be synthesized into concise main text panels with minimal supplementary data.
- 4) Statistics need to be provided for the septa/cell data throughout. For example, in figure S6, I believe that statistical analysis will show that the Δ glpR mutant strain contains more septa per cell. This is not the conclusion in the text. Why?
- 5) The order of data presentation becomes quite muddled between figures 2 and 5. Reorganization should be considered. For example, Figure 3 could be combined with 4. S5 should be integrated into the main text. 4c should be moved to figure 2.
- 6) The Wag31 band in 5h should be indicated.
- 7) Not all figure panels are clearly referred to in the text (e.g. 5e and f)
- 8) Remove the reference to Figure S8 in the discussion. This is not an adequately controlled experiment, and the data contradict lines 334-336.
- 9) Figure 5H. Why is glpR/wag31 interaction reduced in glp knockout? Perhaps GlpP is not located at the septum in the absence of Glp? If this is the hypothesis, the authors should localize GlpR in the Glp mutant.
- 10) The authors might reconsider the implications of the ethambutol sensitive phenotype, as this only very indirectly supports the Wag31-related aspects of their model, and may more directly implicate effects of Glp on cell wall metabolism

Author Rebuttal to Initial comments

Referee #1: cell division, Corynebacteriales, cell wall biology

Referee #2: structural biology, bacterial cell biology

Referee #3: cell division, cell wall biology, genetics

Reviewer Comments:

Reviewer #1 (Remarks to the Author):

7The present article presents a comprehensive analysis of the coordination of cell division and polar elongation in *Corynebacteria*, a group of gram-positive bacteria that are known to cause various types of infections. The authors have identified two new members of the divisome, a complex of proteins that are essential for the proper segregation of chromosomes during cell division, in these organisms. The discovery of these genes is a significant achievement, as it represents the first time that such genes have been studied in *Corynebacteria*. This research could have far-reaching implications in the development of novel anti-infective agents against related pathogens, as understanding the molecular mechanisms of cell division and elongation in these organisms could lead to the identification of new drug targets.

We would like to thank the reviewer for this positive assessment of our work and for stressing the importance of discovering new members of the corynebacterial divisome for direct fundamental implications but also important down-stream applications in drug development.

One of the key findings of the study is that a mutant strain lacking either GLP or GLPR, two of the newly identified divisome members, are viable. This suggests that these genes may not be essential for cell division or cell elongation, and could potentially have redundant functions.

Essentiality and viability are complex issues in microbiology, as many genes that are not essential in lab conditions may well be essential in the natural life cycle of the organism, or mutant strains that are viable with a severely affected physiology are likely not viable in the long run. In our work we describe that both *glp* (when deleted) and *glpr* (when overexpressed) have a strong cell division phenotype. Unlike the wild-type, the Δglp strain is not viable when grown in the presence of the cell wall targeting drug ethambutol, indicating that *glp* is essential in these conditions. Overexpression of *glpr*-mNeon is mostly not tolerated by the bacteria, as most bacteria lose the plasmid and the strongly branched cells will lyse eventually. If, as we think, GLP and GLPR are involved in regulation of the divisome-elongasome transition, we would not expect these proteins to be absolutely indispensable (as it is the case for FtsZ). However, we would expect them to interfere with normal cell cycle progression, and we showed that this is the case for Δglp . In conclusion, both proteins may not be fully essential under certain growth conditions but are required for correct cell cycle progression.

Additionally, this implies that other proteins may also be involved in this process. The authors have excluded two other MoeA-like proteins, but it is important to investigate if there are any other candidate proteins in the immunoproteomics data that could have a similar role.

We definitively expect other candidates to be part of the fully functional divisome, as each building block will have a role in correct cell cycle progression. Whether these are individual pieces each adding to correct function or whether they carry out redundant functions is impossible to say without full mechanistic comprehension of each of the candidates and the complete characterization of the division and elongation machineries, something that we are still very far from understanding, especially in *Corynebacteriales*. However, redundancy remains a possibility and we have now mentioned this possibility in the main text (line 221)

Another noteworthy finding is that a delta GLP mutant can continue to elongate. This has major implications in the analysis of the data, as it suggests that the lack of these coordinators does not have a major effect on elongation or cell division. This, in turn, suggests that the role of GLP/GLPR in these processes is secondary and could be covered to some extent by other proteins. The authors should therefore provide more detailed analysis of this finding and the possible mechanisms underlying this observation.

In *Corynebacteriales* elongation occurs at the poles and is spatially separated from the division site, except for the precise time point during the cell cycle when the septum becomes a new pole (at cytokinesis). The two proteins identified in this work, GLP and GLPR, exclusively localize to the septum and not to the poles and are therefore not expected to interfere with polar elongation. Our proposal is that they are involved in the septum to pole transition-via the direct interaction with FtsZ and Wag31. In the Δglp strain, the cells are multiseptated which means that something goes wrong with finalizing division. This is where we think GLP-GLPR act.

Finally, the authors should provide growth curves to compare the effect of deleting GLP or GLPR in the replication of *Corynebacterium glutamicum*, as this would provide a more comprehensive evaluation of the impact of these genes on the divisome and elongasome of *Corynebacteria*.

We have done the growth curves (see below, left), but they do not differ from WT, as expected because the cells continue growing. It is only by looking at the cells under the microscope that we see that the Δglp strain grows very differently, with a very strong cell division phenotype. They also differ in their sedimentation properties (see below, right panel), possibly due to cell wall defects.

Overall, the study presents a valuable contribution to the field of microbiology, and it is important for the authors to address the aforementioned points in order to further strengthen their findings.

We would like to thank the reviewer for their input and positive comments on the manuscript and hope to have adequately discussed the concerns raised above.

Reviewer #2 (Remarks to the Author):

In this paper, Martinez et al. identified two proteins involved in cell division in *Corynebacteriales* – GLP and GLPR that bear resemblance to eukaryotic gephyrin-like protein and its receptor respectively. Using biochemical and structural approaches, the authors show that GLP binds to the C-terminal of the major bacterial cell division protein, FtsZ and bridges the interaction between the septal cell division machinery and the polar cell elongation machinery via interaction of GLPR with Wag31, the major component of the polar elongosome in *Corynebacteriales*. It is a well-written paper for a broad audience and shows an important advance in our understanding of the less-common lab bacteria belonging to *Corynebacteriales*, especially how the activities of components of the septal divisome and the polar elongosome are coordinated in space and time. Parts of the paper which need more explanation and clarity in my opinion are their data on interaction of SepF/GLP which I have mentioned in my report below.

We would like to thank the reviewer for their positive evaluation of our work and we hope to have clarified the concerns addressed below.

1. Line 58 in Introduction – the authors mention that so far SepF is the only direct interactor of FtsZ in actinobacteria. Recently, Ramos-Leon et al. in 2021 also found SepH in *Streptomyces* and *Mycobacteria*

10that interacts with and regulates FtsZ dynamics. Perhaps, worth including the information as well to keep the readers upto date with the literature.

Thank you for pointing this out. In line 58 we were specifically referring to the direct interaction with the FtsZ-CTD, which to our knowledge has only been proven to be the case for SepF. However we did identify SepH in our interactomes and we have now highlighted SepH in Figure 1a as well as in the main text referring to the work by Ramos-Leon et al. (lines 117-118).

2. The paper is already very well and clearly written for a broad audience. This is only a recommendation, so feel free to take it or not. I feel in the introduction, it may be useful to explain a little bit more about the polar growth in actinobacteria vs lateral growth in more commonly studied bacteria like E. coli or B. subtilis, even if it is just a few sentences of a primer. This will be very useful for microbiologists who do not work with actinobacteria and help them appreciate as to why the cell division mechanisms are starkly different in actinobacteria compared to other commonly studied bacteria. The authors do mention about the polar growth a little bit but explaining and differentiating it from lateral growth will help the readers appreciate the significance of their findings even better, in my opinion.

We have now added a short description of lateral elongation in the introduction (lines 67-69).

3. Any particular reason why the authors didn't use co-IP using FtsZ as a bait directly and used SepF instead?

The reasons for this are multiple. First, we wanted to find early divisome components and we knew from our previous work (Sogues et al Nat Comm 2020) that SepF and FtsZ are involved together in the initial steps of Z-ring assembly. Moreover, we had already characterized several mutants of SepF, which allowed us to do differential interactomics without directly touching FtsZ, but where we were sure to pull down FtsZ and any additional early divisome partners. Also, by pulling on SepF we had FtsZ as a positive control, which was important to optimize our cross-linking protocol to assure that we were able to bring down spatially assembled pieces of the divisome.

4. Fig 1b – for the graph showing the no. of septa, it will be nice to have a legend within the figure itself for the Cglu and Cglu_Δglp strain. Also, may be make the y-axis “percent of cells”

We have added these changes to all the figures that show septal counts, together with the additional suggestions from reviewer #3 (point 4). The corresponding Figures are now Figures 1c, 4f, S2i, S4b.

5. Fig 1e and 5a – it is not exactly clear what the heatmap represents. Is the intensity corresponding to the no of cells with that particular localization on the cell? Will be great to explain in methods or text somewhere.

Heatmaps represent the averaged fluorescent signal with the corresponding averaged cell contour of the detected cells in a dataset. Briefly, each individual point of the cell contour was converted into relative coordinates along the medial axis of the bacterial cell using a polar coordinate system. The resulting representations of the cell contours can be easily averaged into a comprehensive representation of all the cell shapes into a dataset. For each cell, the fluorescent signal was extracted and straightened using their respective medial axis, and then transformed using a 2D affine model to fit with the averaged contour. The resulting images were then averaged to produce the heatmap.

We have added a sentence to the Materials section (lines 784-785) to explain this.

6. As per the results in Fig 1b, Cglu_ Δglp cells, in addition to being longer due to incomplete division, are also wider and the authors notice excessive accumulation of Wag31 in Cglu_ Δglp cells in Fig 5g. Can the authors speculate on why the cells become wider? Is there any evidence linking dysregulation of polar elongosome assembly in Cglu_ Δglp cells with perhaps Wag31 or other elongosome components participating in lateral cell wall build up or is it due to accumulation of lateral cell wall enzymes?

An excess accumulation of Wag31 seems indeed to be associated with wider poles and this can be seen from time to time in cells that greatly overexpress Wag31, where eventually one pole becomes huge (see picture below). Why this is the case is not understood. One possibility could be that Wag31 accumulation by itself can lead to larger poles (maybe by deforming the membrane?) or else it may be excessive Wag31 accumulation that leads to aberrant peptidoglycan synthesis and/or remodeling as Wag31 is thought to assemble the elongosome; more likely it could be a combination of both. Another possibility could be that as in the multiseptate cells the divided compartments become smaller in length, the cell may compensate by becoming wider. We have now mentioned the difference in cell width (line 126).

Cglu_Wag31-mNeon phase contrast and mNeon signal overlaid; scale bar 5μm

12

This content is licensed under a Creative Commons Attribution 4.0 International License, which permits use, sharing, adaptation, distribution and reproduction in any medium or format, as long as you give appropriate credit to the original author(s) and the source, provide a link to the Creative Commons license, and indicate if changes were made. In the cases where the authors are anonymous, such as is the case for the reports of anonymous peer reviewers, author attribution should be to 'Anonymous Referee' followed by a clear attribution to the source work. The images or other third party material in this file are included in the article's Creative Commons license, unless indicated otherwise in a credit line to the material. If material is not included in the article's Creative Commons license and your intended use is not permitted by statutory regulation or exceeds the permitted use, you will need to obtain permission directly from the copyright holder. To view a copy of this license, visit <http://creativecommons.org/licenses/by/4.0/>.

7. Line 169 - It is kind of surprising that the authors could not detect direct binding between SepF and GLP which contradicts their Fig 1a interactomics data. Why do the authors think they could not detect direct interaction between SepF and GLP *in vitro* – anything to do with the conditions they used? What about in the presence of FtsZ? Can SepF and GLP interact *in vivo* in the presence of FtsZ? Did the authors consider doing a two-hybrid assay for the same?

There is no contradiction, because Figure 1 does not refer to physical direct protein-protein interaction but to interaction networks or enriched components of SepF-mediated complexes (including both direct and indirect interactors). We have added a note in the legend of Figure 1a to make this clearer and added a note to lines 111-112 in the main text. Given that we cross-link our cells (to preserve the architecture of the septum, including FtsZ polymers and membrane organization), we expect some of the interactors to be indirect. And even without cross-linking, immunoprecipitation coupled to MS experiments can never distinguish direct from indirect interactors. This is one of the reasons we included the SepF mutant that cannot bind FtsZ, as this should give a differential interactome for all the interactors that occur through FtsZ (Figure 2a).

In order to detect interactions *in vivo*, we preferred to use an MS-based proteomics approach rather than two-hybrid assays. Although both methods will detect direct and indirect partners, we believe the former method is more unbiased and less prone-to-artefacts. In fact, two-hybrid assays (even in *E. coli*, where neither SepF, GLP or GLPR are present) can lead to indirect readouts because the FtsZ-CTD (that directly interacts with SepF and GLP) is highly similar between *E. coli* and *Cglu*.

And *in vitro*, since we were able to probe direct protein-protein interactions using biophysical methods with purified recombinant proteins, two hybrid assays were not necessary.

8. Fig 2a – what do these normalized XIC intensity values mean? If we look at the absolute nos, the XIC

intensity in case of FtsZ/SepF mutant is like SepF WT/GLP interaction. Does it mean that the SepF WT/GLP interaction is weak to begin with?

XIC refers to extracted ion chromatogram and is one of the methods routinely used to obtain label-free quantitative data from nanoLC-MS/MS datasets. Briefly, for each peptide the intensity of the signal (for its m/z value) is extracted from a series of mass spectra (MS level) acquired during chromatographic separation and the XIC intensity of each protein can be computed using specific software. We normalized the XIC of each protein with the XIC of the bait to correct for small changes in bait recovery in the different replicates. In summary XIC is a measure of protein abundance based on its “intensity”. This value can be used to compare the same protein along different conditions, but as each peptide sequence has a different mass spectrometry response (giving rise to differences in intensities not related to the amount of peptide but to its physicochemical properties) it is not useful for comparisons of different proteins. Thus, the FtsZ/SepF ratio cannot be compared to the SepF/GLP. But when we compared the GLP/SepF ratio in different contexts (wt and using a mutant that is impaired in FtsZ binding) we can conclude that FtsZ is important for the GLP-SepF direct/indirect interaction *in vivo*. We have now included a sentence in materials and methods (**lines 732-733**).

9. Did the authors look at the localization of mNeon-GLP in SepF mutant? It will be interesting to see that and will further bolster the claim about interaction of GLP with FtsZ.

We had done this experiment (see our original Supplementary Figure S8), but it is not as informative as we could have expected. We observed that in the SepF depletion strain, GLP becomes more cytosolic and mostly delocalized but sometimes accumulates at septal vestiges (probably prior to depletion) and interestingly can go to poles which in normal conditions is not the case. We believe that this is due to the absence of a functional divisome where FtsZ prevents GLP from going to the poles. However, following the suggestion of reviewer #3 (see point 8 below), we have now removed this figure in the revised version, as it is not possible to quantify localization in these very heterogenous and sick cells.

10. Based on their data and model, is SepF even then necessary for the interaction of FtsZ and GLP? I am kind of left wondering about the chronology of steps for recruitment of cell division proteins. Does FtsZ recruit SepF first and then GLP or are they recruited independently to FtsZ C-terminal domain? Even in their model in Fig 6b/6a, there doesn't seem to be any interaction b/w SepF and GLP/GLPR complex as per their schematics, then how come GLP was detected in the interactome of SepF? Am I missing any method details or other justification here?

We think there is a confusion between the interactome shown in Figure 1, which includes both direct and indirect interactions (see our answer to point 7 above), and the interaction network shown in Figure 6a, which only includes direct protein-protein interactions (validated *in vitro*). In the manuscript, we showed that GLP directly binds FtsZ (i.e., independently of SepF) using different approaches: biolayer interferometry (Figure 2b), thermofluor (Figure S2b) and through structural analysis of the GLP-FtsZ complex by X-ray crystallography (Figure 2c-d).

As for the sequence of events, we cannot tell precisely at which moment GLP interacts with FtsZ. We know from our previous work (Sogues et al Nat Comm 2020) that SepF and FtsZ are interdependent to go to the future site of septation, where they arrive together, and this is a prerequisite for Z-ring assembly and septum formation. For GLP we know that it arrives very early to the site of division (Figure 1e, white arrow), because mNeon-GLP can be seen before the membrane (septum) formation. From our results, we conclude that FtsZ contributes to early GLP recruitment, suggesting that FtsZ in *Corynebacteriales* could act as a hub for protein-protein interactions at the septum, something that was already known for other bacterial models.

11. Minor point – line 368 – the reference is in a different format from the rest of the article.

Thank you for pointing this out we have corrected the reference format.

Reviewer #3 (Remarks to the Author):

The work by Petit et al. investigates the structure and regulation of the actinobacterial divisome. This is an important topic, as it is quite clear that this phylum of bacteria, which includes major human pathogens and organisms important for biotechnology, grows and divides using mechanisms that are different from well-studied model organisms. The authors use an impressive combination of genetics, biochemistry, structural biology, and cell biology to propose the presence of a novel protein complex that regulates the machinery necessary for septation and elongation. While interesting and clear conclusions can be drawn from this work, a number of important aspects of model are supported by tenuous evidence, making the overall model quite speculative. The major claims of the manuscript are summarized below:

1) A previously uncharacterized protein, named Glp, is a divisome component that is important for septation in *C. glutamicum*. This claim is well supported by genetic and biochemical evidence.

2) Glp is paralogous to MoaA, indicating that this protein has evolved a new function in actinobacteria that is unrelated to its annotated role in MoCo synthesis. This is interesting and well supported by phylogenetic studies (though the presentation of this work could be greatly improved).

153) Glp forms a complex with the co-operonic protein, GlpR. This is supported by pulldowns (after crosslinking) and in vitro biochemical assays. The importance of this interaction in vivo could be further supported by using existing reagents to investigate whether Glp and GlpR depend on each other for septal localization.

4) The Glp/GlpR complex regulates septation and/or elongation. This conclusion is supported by reasonably strong, though indirect evidence. A role for GlpR in controlling cellular morphology is demonstrated upon overexpression of a Neon-tagged version of the protein or a mutant protein lacking the GlpR binding site, but deletion of the gene or expression of the native protein has no effect. It's understandable that some redundancy may obscure a phenotype in the knockout, and the evidence obtained by overexpression of the binding site mutant is reasonably compelling.

5) GlpR regulates cell elongation via binding to Wag31. While the biochemical data supporting this interaction is adequate, no convincing evidence supports a function for this interaction in the cell. The authors claim that Wag31 "accumulates" at the septum in Δ glp cells, but there is no control to show normal levels of this protein at the septum. Further, they suggest that observing Wag31 at sites of branching in the GlpR-Neon overexpression strain is evidence for functional interplay between the proteins, but Wag31 will always be found at sites of septation/branching. As a result, both this claim and the model presented in Figure 6 are very speculative.

We would like to thank reviewer #3 for a very sharp and accurate analysis of our manuscript. We fully agree with the comments and positive criticism, and we have made changes accordingly. In particular, we are aware that the final part of the manuscript is speculative, as the link to the elongasome is particularly difficult to address from a mechanistic point of view. We have now toned down, or plainly removed, some of the original claims linked to the role of the GLP/GLPR complex in Wag31 accumulation and elongasome maturation. Nevertheless, we think that the biochemical/biophysical evidence for the direct binding of Wag31 to GLPR is an important result, because Wag31, just like DivIVA, is a notoriously difficult protein to work with in solution, which has hampered for many years any significant progress on our mechanistic understanding of this protein. Indeed, to the best of our knowledge, GLPR binding represents the first proof for direct, *in vitro*, protein-protein interactions involving Wag31/DivIVA, since many of the previous interactors described in the literature come from genetic or cellular studies. So even if the cellular data will require much more work to understand the precise functional role of this system with regard to the elongasome, the fact that these proteins interact is important per se, and paves the way for additional molecular studies on this untamable protein.

A valuable manuscript could be crafted based on only the first 4 points, concluding that Glp/GlpR contribute to septation in some important but still unclear manner.

If the authors wish to make the additional broad mechanistic claims related to Wag31, a significant amount of work would be necessary and it's unclear what the outcome would be. Either way, the manuscript is not written or organized in a manner that is sufficient for publication, and the following points should be addressed:

We have addressed all the reviewer's concerns below and restructured the manuscript accordingly. We have re-organised the figures as well as the supplementary data and removed the final part from the results (original Figure 5g and S8) and the most speculative part from the discussion (link to Wag31 accumulation and pole formation). We have also redrawn our working model in Figure 6b.

1) Throughout the manuscript, the supplementary data section is used inappropriately, separating essential experimental controls from the main text data. In general, the reader should not be forced to refer to the supplement to interpret main text figures. Specifically, this organization makes it impossible to evaluate the data, as it is unclear if the experimental and control data are from the same experiment (e.g. complemented strains, etc.).

We have restructured some of the supplementary figures and whenever possible and appropriate we have moved the supplementary to the main figures. In particular:

- we have added the complementation data from Supplementary Figure 1 into the main **Figure 1**
- BLI experiments have been represented differently (see also reply to point 3 below)

2) The phylogenetic data in figure 1 is not explained adequately. Furthermore, it seems to rely on synteny between Glp and GlpR, but the latter is not yet introduced. I highly recommend moving this figure and discussion to the end of the manuscript, and explaining the reasoning more thoroughly.

We would like to thank the reviewer for this suggestion. We have now regrouped all the phylogenetic results together with the structure-sequence analysis of the paralogs in a stand-alone paragraph at the end of the results section and prepared a new Figure 5 centered on the evolutionary repurposing and actinobacteria-specific evolution of GLP-GLPR. We agree that this may be a more clear and impactful way of representing the data.

3) The BLI studies seem sound, but they are presented like they were copied from a lab notebook. These panels should be synthesized into concise main text panels with minimal supplementary data.

We have now removed the fitting data from the supplementary, put the values +/- SD into the main text (lines 160, 204, 253) and only show the binding curves in the main figures for the protein-protein interactions between GLP-FtsZ (Figure 2b), GLP-GLPR (Figure 3b) and GLPR-Wag31 (Figure 4i). We have kept the controls that validate the binding (ie showing a non-binding event in the GLP_{Δloop} mutant) in the supplementary figures (S2g and S3). The additional binding between GLPR and Wag31₁₋₆₁ is also shown in Supplementary Figure S5. These latter supplementary figures are in support of the main findings, but we believe they are not required to understand the paper as such. We hope to have addressed the aesthetics question around these representations to the satisfaction of the reviewer.

4) Statistics need to be provided for the septa/cell data throughout. For example, in figure S6, I believe that statistical analysis will show that the ΔglpR mutant strain contains more septa per cell. This is not the conclusion in the text. Why?

We have now changed all the septal count histograms to make them easier to “read”. For this we have pooled the 0/1 categories, as these can be considered normal WT-like cell-cycle states. We grouped the analysis per strain as this makes the statistical analysis more relevant (comparisons done on a population level). We have added statistical significance whenever appropriate and modified the legends accordingly. These changes can now be found in the corresponding Figures 1c, 4f, S2i, S4b.

In the histogram in the original Figure S6 (now S4) that represents the number of septa per cell there is indeed a difference in the zero and one septum per cell. However, both of these conditions can be considered wild-type like. If we pool the 0/1 into one group to represent the population with a normal septal count the difference is largely reduced (not significant). *Cglu_ΔglpR* (in blue) presents hardly any multiseptated cells.

5) The order of data presentation becomes quite muddled between figures 2 and 5. Reorganization should be considered. For example, Figure 3 could be combined with 4. S5 should be integrated into the main text. 4c should be moved to figure 2.

We have considerably changed the Figures to be in line with the comments and suggestions provided. In summary, we have merged the original Figure panels 1e, 2d, 2e, and 3 into one new Figure 5 describing the evolutionary repurposing of GLP and phylogeny of GLP-GLPR. The text has been changed accordingly and all this information is now in one separate paragraph (lines 260 to 287). We moved Figure 4c to Figure S2e to avoid confusion between FtsZ binding to GLP and GlyR binding to Gephyrin. The main Figure 2 and S2 now represent all the structural analysis of GLP and GLP-FtsZ. The BLI data has been changed as described above. In particular, the fitting data of Figure S5a has been removed

and values described in the main text. We kept Figure S5b (now S3) as this shows a non-binding event and the best way to represent it is by comparing the binding profiles.

6) The Wag31 band in 5h should be indicated.

In addition to the arrows we have labelled Wag31 in the elution fraction by a pink star in what is now Figure 4g.

7) Not all figure panels are clearly referred to in the text (e.g. 5e and f)

Thank you for pointing this out, we have now referred to these in the results section under Figure 4e and 4f (lines 230, 232).

8) Remove the reference to Figure S8 in the discussion. This is not an adequately controlled experiment, and the data contradict lines 334-336.

We agree with the reviewer and have removed this figure (also in line with toning down the more speculative part 5) mentioned in the Summary.

9) Figure 5H. Why is *glpr/wag31* interaction reduced in *glp* knockout? Perhaps GlpP is not located at the septum in the absence of Glp? If this is the hypothesis, the authors should localize GlpR in the Glp mutant.

We have verified this, and indeed, in the absence of GLP, the GLPR protein levels are reduced in the cell (shown in new Figure S10c (Antibody characterization)). This suggests that GLP could be required for GLPR stability *in vivo*, possibly by preventing the large intrinsically disordered intracellular regions to be exposed to proteolysis. Alternatively, it could also be due to changes at the transcriptional level in the absence of GLP.

We have verified the localization of GLP in *Cglu_Δglpr* (new Figure 3d) described in the main text (lines 212-213). We also verified GLPR-mNeon in the *Cglu_Δglp* background (shown below). Both proteins can localize in the absence of the other. This would be expected as GLP can also bind FtsZ and GLPR can bind Wag31.

GLPR-mNeon localizes to the multiple septa of the *Cglu_Aglp* strain. For the heatmap, cells have been sorted by number of septa, indicated below the maps.

10) The authors might reconsider the implications of the ethambutol sensitive phenotype, as this only very indirectly supports the Wag31-related aspects of their model, and may more directly implicate effects of Glp on cell wall metabolism

We agree with the reviewer that the ethambutol sensitivity is a very indirect measure of a possible functional implications in division or elongation, and we have therefore toned down this discussion and also mentioned more direct effects on cell wall metabolism as an alternative explanation for this phenotype (lines 355-358).

Decision Letter, first revision:

Message: Our ref: NMICROBIOL-22123129A

12th July 2023

Dear Dr. Wehenkel,

Thank you for submitting your revised manuscript "Eukaryotic-like gephyrin and cognate membrane receptor coordinate corynebacterial cell division and polar elongation." (NMICROBIOL-22123129A). It has now been seen by the original referees and their comments are below. The reviewers find that the paper has improved in revision, and therefore we'll be happy in principle to publish it in Nature Microbiology, pending minor revisions to satisfy the referees' final requests and to comply with our editorial and formatting guidelines.

We are now performing detailed checks on your paper and will send you a checklist detailing our editorial and formatting requirements in about a week. Please do not upload the final materials and make any revisions until you receive this additional information

20from us.

Thank you again for your interest in Nature Microbiology Please do not hesitate to contact me if you have any questions.

Sincerely,

Kyle

Dr. Kyle Frischkorn
(he/him/his)
Senior Editor, Nature Microbiology
Nature Portfolio

Reviewer #1 (Remarks to the Author):

I have no further comments, the authors have fully addressed all my concerns and the manuscript is ready for publication.

Reviewer #2 (Remarks to the Author):

Thank you for addressing all the points in detail and for your work on the manuscript. It should be a good article for the cell division field!

Reviewer #3 (Remarks to the Author):

The revised manuscript is greatly improved – it is now clear, logical, and easy to evaluate without referring to the supplement. In addition, modifications to the Wag31-related conclusions significantly increases the rigor of the work. After a couple of small text edits (below), the manuscript will represent a valuable contribution to the field.

- 1) Line 130 – the authors should explicitly state that the cell length and width phenotypes of the Δ glp mutant are not complemented. Obviously, these are phenotypes are not rigorously attributed to Glp, and this should be clear to the reader.
- 2) Throughout, conventions for bacterial protein name designations should be consistent: “Glp” not “GLP”.

3) Specific Wag31-related conclusions in the main text are well-supported. However, in my opinion, some of the text related to these findings in the abstract and introduction remains overstated. The authors should consider whether their work really supports the following:

Abstract: “...that the interplay between the GLPR/GLP module, FtsZ, and Wag31 is crucial for orchestrating cell cycle progression. Our results provide a detailed molecular understanding of the crosstalk between... the divisome and elongasome”.

Introduction: “GLP...interferes with cell elongation via direct interactions with GLPR and Wag31”. This sentence seems particularly overstated, as the functional effect of Glp/GlpR on Wag31 remains unclear.

Author Rebuttal, first revision:

Second round of comments from reviewers:

Reviewer #1:

Remarks to the Author:

I have no further comments, the authors have fully addressed all my concerns and the manuscript is ready for publication.

Thank you for your time and advice that allowed us to improve the manuscript.

Reviewer #2:

Remarks to the Author:

Thank you for addressing all the points in detail and for your work on the manuscript. It should be a good article for the cell division field!

Thank you for your time and advice that allowed us to improve the manuscript.

Reviewer #3:

Remarks to the Author:

The revised manuscript is greatly improved – it is now clear, logical, and easy to evaluate without referring to the supplement. In addition, modifications to the Wag31-related conclusions significantly increases the rigor of the work. After a couple of small text edits (below), the manuscript will represent a valuable contribution to the field.

We would like to thank the reviewer for these positive remarks, and we have added below the few remaining changes suggested.

1) Line 130 – the authors should explicitly state that the cell length and width phenotypes of the Δ glp mutant are not complemented. Obviously, these are phenotypes are not rigorously attributed to Glp, and this should be clear to the reader.

We are not sure about the point made here, or if there is a confusion in the line numbering. In Figure 1 we show that the Δ glp mutant is fully complemented both with the tagged and non-tagged version of Glp and that the differences between the WT and these complemented mutants are statistically non-significant.

2) Throughout, conventions for bacterial protein name designations should be consistent: “Glp” not “GLP”.

22Ok we have now changed GLP to Glp and GLPR to GlpR, in analogy to the GlyR nomenclature.

3) Specific Wag31-related conclusions in the main text are well-supported. However, in my opinion, some of the text related to these findings in the abstract and introduction remains overstated. The authors should consider whether their work really supports the following:

Abstract: "...that the interplay between the GLPR/GLP module, FtsZ, and Wag31 is crucial for orchestrating cell cycle progression. Our results provide a detailed molecular understanding of the crosstalk between... the divisome and elongasome".Introduction: "GLP...interferes with cell elongation via direct interactions with GLPR and Wag31". This sentence seems particularly overstated, as the functional effect of Glp/GlpR on Wag31 remains unclear.

We have modified the abstract and introduction to tone-down these claims.

Final Decision Letter:

Mes 11th August 2023

sag

e: Dear Anne Marie,

I am pleased to accept your Article "Eukaryotic-like gephyrin and cognate membrane receptor coordinate corynebacterial cell division and polar elongation." for publication in Nature Microbiology. Thank you for having chosen to submit your work to us and many congratulations.

Acceptance of your manuscript is conditional on all authors' agreement with our publication policies (see <https://www.nature.com/nmicrobiol/editorial-policies>). In particular your manuscript must not be published elsewhere and there must be no announcement of the work to any media outlet until the publication date (the day on which it is uploaded onto our website).

Please note that *Nature Microbiology* is a Transformative Journal (TJ). Authors may publish their research with us through the traditional subscription access route or make their

1paper immediately open access through payment of an article-processing charge (APC). Authors will not be required to make a final decision about access to their article until it has been accepted. [Find out more about Transformative Journals](https://www.springernature.com/gp/open-research/transformative-journals)

Authors may need to take specific actions to achieve [compliance](https://www.springernature.com/gp/open-research/funding/policy-compliance-faqs) with funder and institutional open access mandates. If your research is supported by a funder that requires immediate open access (e.g. according to [Plan S principles](https://www.springernature.com/gp/open-research/plan-s-compliance)) then you should select the gold OA route, and we will direct you to the compliant route where possible. For authors selecting the subscription publication route, the journal's standard licensing terms will need to be accepted, including [self-archiving policies](https://www.nature.com/nature-portfolio/editorial-policies/self-archiving-and-license-to-publish). Those licensing terms will supersede any other terms that the author or any third party may assert apply to any version of the manuscript.

As soon as your article is published, you will receive an automated email with your shareable

link.

With kind regards,

[redacted]

P.S. Click on the following link if you would like to recommend Nature Microbiology to your librarian <http://www.nature.com/subscriptions/recommend.html#forms>

** Visit the Springer Nature Editorial and Publishing website at http://editorial-jobs.springernature.com?utm_source=ejp_NMicro_email&utm_medium=ejp_NMicro_email&utm_campaign=ejp_NMicro for more information about our career opportunities. If you have any questions please click [here](mailto:editorial.publishing.jobs@springernature.com).**